# Conditional knockout of *Shank3* in the ventral CA1 by quantitative in vivo genome-editing impairs social memory in mice

Myung Chung [1,2], Katsutoshi Imanaka[1,2], Ziyan Huang [1,2], Akiyuki Watarai[1], Mu-Yun Wang [1], Kentaro Tao [1], Hirotaka Ejima [3], Tomomi Aida [4], Guoping Feng [4,5] & Teruhiro Okuyama [1,2] ✉

Individuals with autism spectrum disorder (ASD) have a higher prevalence of social memory impairment. A series of our previous studies revealed that hippocampal ventral CA1 (vCA1) neurons possess social memory engram and that the neurophysiological representation of social memory in the vCA1 neurons is disrupted in ASD-associated *Shank3* knockout mice. However, whether the dysfunction of Shank3 in vCA1 causes the social memory impairment observed in ASD remains unclear. In this study, we found that vCA1-specific *Shank3* conditional knockout (cKO) by the adeno-associated virus (AAV)- or specialized extracellular vesicle (EV)- mediated in vivo gene editing was sufficient to recapitulate the social memory impairment in male mice. Furthermore, the utilization of EV-mediated *Shank3*-cKO allowed us to quantitatively examine the role of Shank3 in social memory. Our results suggested that there is a certain threshold for the proportion of *Shank3*-cKO neurons required for social memory disruption. Thus, our study provides insight into the population coding of social memory in vCA1, as well as the pathological mechanisms underlying social memory impairment in ASD.

Autism spectrum disorder (ASD) is a heterogeneous neurodevelopmental disorder characterized by persistent deficits in social communication in conjunction with highly restricted, stereotyped, repetitive behaviors[1]. Individuals with ASD have a higher prevalence of comorbid neuropsychiatric conditions such as intellectual disability, depression, attention-deficit hyperactivity disorder (ADHD), anxiety disorder, and social memory impairment[2]. The latest estimated heritability of ASD is 80.8%, based on population-based cohort studies, which have provided strong evidence of genetic contribution to ASD[3]. Recent genome-wide association studies (GWAS) have revealed that mutations in various genes essential for synapse formation and synaptic function are associated with ASD, which has led many researchers to classify the disorder as a "Synaptopathy"[4–6]. Among these genes, SH3

and multiple ankyrin repeat domains 3 (Shank3), which encodes a glutamatergic postsynaptic scaffolding protein, is one of the most promising ASD candidate genes[7,8]. Mutations in the *Shank3* gene comprise approximately 1% of the idiopathic ASD population, which makes it one of the most prevalent monogenic causes of ASD[8–11]. The Shank3 protein is expressed in multiple brain regions, including the cortex, striatum, thalamus, and hippocampus[9]. Previous studies have reported reduced spine density, morphological abnormalities in postsynaptic density, and perturbed postsynaptic function in the striatum of *Shank3* knockout (*Shank3*-KO) mice[12]. Similar morphological and functional abnormalities were observed in the anterior cingulate cortex (ACC)[13]. Optogenetic manipulation using *Shank3*-KO mice and region-specific perturbation and restoration experiments of

[1]Laboratory of Behavioral Neuroscience, Institute for Quantitative Biosciences, The University of Tokyo, Tokyo, Japan. [2]Graduate School of Medicine, The University of Tokyo, Tokyo, Japan. [3]Department of Materials Engineering, School of Engineering, The University of Tokyo, Tokyo, Japan. [4]McGovern Institute for Brain Research, Department of Brain and Cognitive Sciences, Massachusetts Institute of Technology, Cambridge, MA, USA. [5]Stanley Center for Psychiatric Research, Broad Institute of MIT and Harvard, Cambridge, MA, USA. ✉e-mail: okuyama@iqb.u-tokyo.ac.jp

*Shank3* gene demonstrated the contribution of Shank3 in repetitive behavior, hypersensitivity, and sociability in the striatum, somatosensory cortex, and ACC, respectively[9,12–15]. In contrast, how the dysfunction of the *Shank3* is involved in comorbidities in ASD, including social memory impairment, has not been examined.

Social memory, the ability to remember and recognize other conspecific individuals, is indispensable for social animals including humans to exhibit appropriate social communication with conspecifics in their social group[16]. Recent studies have shown that the ventral CA1 (vCA1) and dorsal CA2 (dCA2) subregions of the hippocampus are essential for social memory[17–19]. In particular, dCA2 pyramidal neurons process social memory information by receiving and conveying social information via neuropeptide modulation[19–23], whereas vCA1 pyramidal neurons store social memories by receiving information from dCA2 neurons[18,21,24–26]. Disrupted social memory has been reported in individuals with ASD without intellectual disabilities and ASD model animals[27,28]. Furthermore, our research group recently reported that the proportion of vCA1 social memory neurons was reduced and the temporal coding of the neural ensemble during social exploratory behavior and sharp-wave ripples (SPW-Rs) during the sleep period were disrupted in *Shank3*-KO mice[25]. Despite growing evidence for the contribution of vCA1 to the pathological features of ASD, it remains unclear whether vCA1 is the region responsible for social memory impairment in ASD.

Recently developed gene editing systems are powerful tools for the functional analysis of target genes and enable the precise manipulation of genomic DNA in living organisms. One of the most reliable approaches for delivering editing molecules to target tissues in vivo is the use of viruses, such as adeno-associated virus (AAV). However, gene delivery using AAV can lead to prolonged expression in infected cells, which increases the frequency of off-target editing and fails to discuss the quantifiability of edited cells. These drawbacks of viral delivery have motivated the development of alternative strategies for delivering in vivo editing molecules. In this study, we used extracellular vesicles (EVs), which are non-viral, rapid, robust, and size-unlimited platforms, to deliver the editing molecules. Non-viral, EV-mediated in vivo genome-editing is gaining increasing attention owing to several advantages over AAV, such as larger cargo capacity, transient expression, better quantifiability, and fewer undesired consequences (e.g., off-target effects, toxicity, and vector integration). However, the in vivo utility of EV-mediated system has not been well explored, particularly in the central nervous system.

Here, we demonstrate that the vCA1-specific conditional knockout (cKO) of *Shank3* through both AAV- and EV-mediated CRISPR/Cas9 system is sufficient to disrupt social memory in mice, suggesting the necessity of Shank3 in the vCA1 region for typical social memory. Using the transient delivery of *Shank3*-targeting Cas9/single guide RNA ribonucleoprotein complexes (RNPs) by EVs in vCA1, serial dilution of EVs suggested that more than a certain threshold of *Shank3* unedited neurons is required to obtain social memory-dependent social discriminatory behavior.

## Results

### *Shank3*-cKO via AAV-mediated CRISPR/Cas9 in vCA1 leads to disrupted social memory

We have previously demonstrated that conventional *Shank3*-KO mice exhibit impaired social discriminatory behavior, with no preference for a novel individual over a familiar one[25]. To examine whether the social amnesia phenotype observed in the conventional *Shank3*-KO mice was caused by the dysfunction of Shank3 in vCA1 neurons, we adopted the AAV-mediated CRISPR/Cas9 method to specifically knockout *Shank3* in vCA1 (Fig. 1a). The mCherry fluorescent protein-coding sequence was inserted after the Cas9 coding sequence to allow the identification of AAV-infected cells, and a single guide RNA (sgRNA) targeting the *Shank3* gene, which has been previously validated[13], was expressed under the *U6* promoter. Next, to assess the social memory of region-specific *Shank3*-cKO-mice, a social discrimination test (Fig. 1b, c) was performed using *Shank3*-targeting sgRNA containing AAV (*Shank3*-cKO-AAV) or blank sgRNA-containing AAV (control-AAV)-injected mice (Fig. 1d, e). We also confirmed the conditional knockout of *Shank3* by immunohistochemistry for staining of the Shank3 protein (Supplementary Fig. 1a). The control-AAV-injected mice spent more time with novel individuals compared to familiar ones, exhibiting normal social memory (Fig. 1f–h, see Supplementary Table 1 for statistical details). In contrast, *Shank3*-cKO-AAV injection into vCA1 resulted in no preference for novel individuals, suggesting that *Shank3* cKO in vCA1 disrupted social memory (Fig. 1g). To compare the groups, we further calculated the social discrimination score, as previously described[18] which showed a reduction in the score by vCA1-*Shank3*-cKO (Fig. 1h).

Next, to examine whether Shank3 functions in social memory in vCA1 downstream neurons, we injected the *Shank3*-cKO-AAV into major afferent projection regions of vCA1 neurons: the ventromedial prefrontal cortex (vmPFC), nucleus accumbens shell (NAcS), and lateral hypothalamus (LH)[29] (Supplementary Fig. 1b–g). The vmPFC-*Shank3*-cKO group and NAcS-*Shank3*-cKO groups showed a preference for novel individuals, suggesting normal social memory, whereas the LH-*Shank3*-cKO group showed disrupted social memory similar to the vCA1-*Shank3*-cKO group (Fig. 1g, h). There was no difference in the total social interaction time during the social discrimination test between the groups (Supplementary Fig. 1h). To investigate whether the *Shank3* expression in the LH neurons, which receive neural projection from vCA1 (^vCA1→LH neurons) is essential for social memory, we performed projection pattern dependent *Shank3* conditional knockout (Supplementary Fig. 2a, b). Cre recombinase was anterogradely derived from vCA1 to the LH by transsynaptic delivery of AAV1-hSyn:Cre, and Cre-dependent Cas9 and sgRNA-expressing AAV were simultaneously injected into the LH (Supplementary Fig. 2a). Social memory dependent behavior was not disrupted by the *Shank3*-cKO in the ^vCA1→LH neurons, which suggests the social memory impairment observed in the *Shank3*-cKO in the LH neurons is not dependent on *Shank3*-cKO in the ^vCA1→LH neurons (Supplementary Fig. 2c, d).

### In vitro confirmation of EV-mediated Cas9/sgRNA ribonucleoprotein complex

Thus far it is unclear what proportion of the *Shank3*-unedited (i.e., normal) vCA1 neurons is required to obtain normal social memory. Therefore, we aimed to determine the proportion of vCA1 neurons with Shank3 dysfunction resulting in the disruption of social memory function. To address this, we introduced a method to apply the CRISPR/Cas9 system to brain tissue in vivo with a specialized EV named "Gesicle"[30]. AAV is widely used to deliver genes of interest to the brain tissue, and the expression of the delivered gene is highly prominent and persists for a long period. Thus, AAV-mediated in vivo genome editing may not be adequate to allow the expression of the delivered gene in a particular cell proportion because of the difficulty in controlling infection and expression. In contrast, cell-derived EVs have been proposed as natural carriers for DNA and other molecules to deliver molecules efficiently in a transient manner and significantly lower the risk of off-target effects[30]. In the EV-mediated system, Cas9 protein was tagged with a transmembrane domain to efficiently incorporate Cas9 protein into EVs, and CherryPicker red fluorescent protein was fused to its extracellular domain to allow fluorescent visualization of the EVs themselves (Fig. 2a). First, we evaluated the genome-editing efficiency of EVs using in vitro cultured cells expressing enhanced green fluorescent protein (EGFP). We extracted EVs from donor HEK293T cells expressing Cas9 and sgRNA targeting the *EGFP* gene and assayed the efficiency of genome-editing when they were introduced into the recipient EGFP reporter stable HEK293 cell line (HEK293-EGFP) (Fig. 2a). We chose four candidate sgRNAs targeting the *EGFP* gene based on the genome editing efficiency

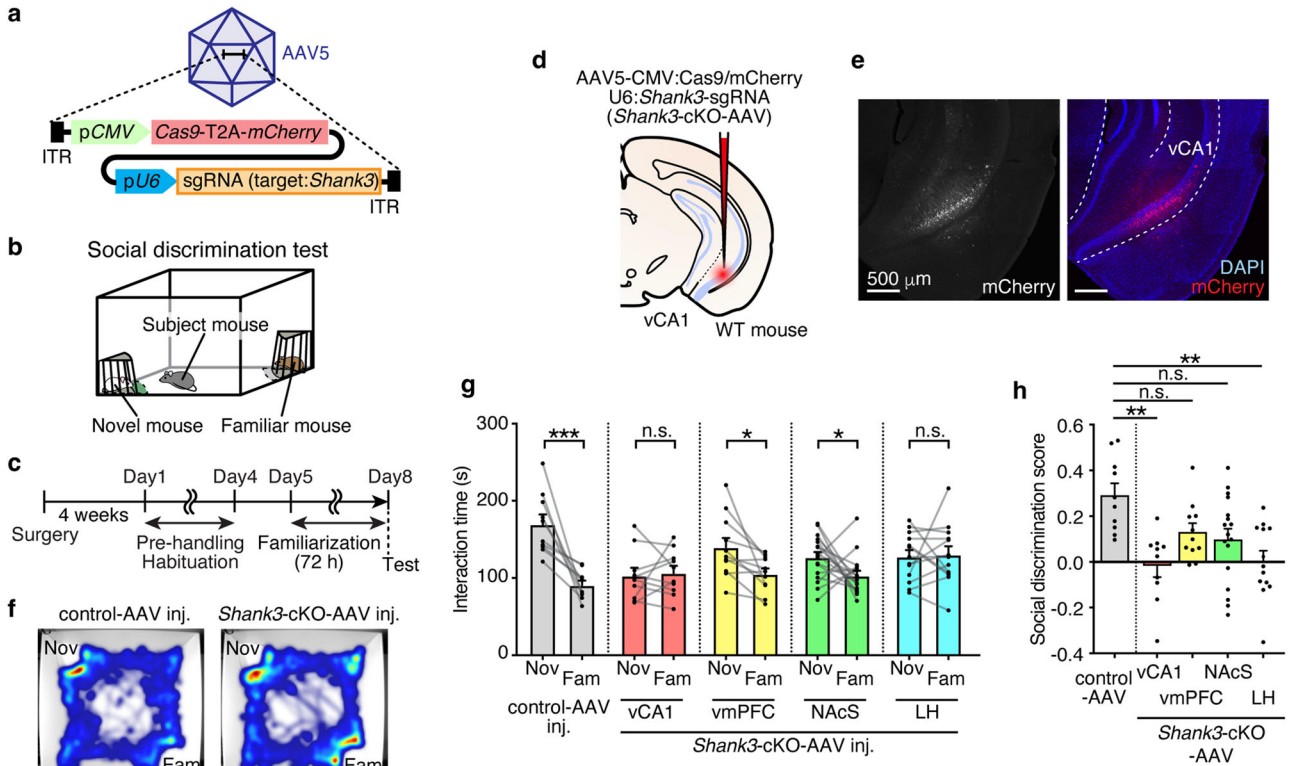

**Fig. 1 | *Shank3*-cKO in the vCA1 using the AAV-mediated CRISPR/Cas9 system impairs social memory. a** Schematic illustration of the *Shank3*-targeting AAV vector. **b, c** Schematic illustration of the social discrimination test chamber (**b**) and schedule (**c**). **d** Schematic illustration of AAV microinjection. **e** Representative confocal microscopy images of AAV injected vHPC stained with anti-RFP (red, for mCherry) and DAPI (blue). Scale bar = 500 μm. **f** Representative heatmap image of social discrimination test. **g** Social discrimination test. (Control AAV, n = 10 mice; vCA1- *Shank3*-cKO, n = 10 mice; vmPFC-*Shank3*-cKO, n = 11 mice; NAcS-*Shank3*-cKO, n = 17 mice; LH-*Shank3*-cKO, n = 13 mice. Two-way mixed model ANOVA). **h** Social discrimination score. (Control AAV, n = 10 mice; vCA1- *Shank3*-cKO, n = 10 mice; vmPFC-*Shank3*-cKO, n = 11 mice; NAcS-*Shank3*-cKO, n = 17 mice; LH-*Shank3*-cKO, n = 13 mice. One-way ANOVA followed by Scheffe's multiple comparisons test). *p < 0.05, **p < 0.01, ***p < 0.001, ****p < 0.0001, n.s. not significant. Data presented as mean ± SEM. See Supplementary Table 1 for exact p values. Source data are provided as a Source Data file.

predicted using in silico genome-editing prediction software[31] (Supplementary Fig. 3a, Fig. 2b). We screened these four sgRNAs to identify the most efficient sgRNA for editing *EGFP* gene. A significant decrease in EGFP fluorescence in CherryPicker-expressing cells was observed three days after EV transduction into HEK293-EGFP recipient cells (Fig. 2c). The percentage of EGFP-expressing cells after EV transduction was quantitatively measured by flow cytometry (Supplementary Fig. 3b, c). While only 3.1% of cells were EGFP negative in the control group, *EGFP*-targeting EVs-#2, #3, and #4 showed significantly increased EGFP negative cells (Supplementary Fig. 3c). Among these three sgRNAs targeting *EGFP* gene, EV-#3 had the highest editing efficiency; therefore, EV-#3 was used in subsequent analyses.

We performed the same assay using diluted EV solutions to examine the correlation between EV concentration and genome-editing efficiency. EVs with varying dilutions (dilution factors: 1/1, 1/2, 1/4, 1/8, and 1/16) and the negative control EVs containing only Cas9 protein were transduced into HEK293-EGFP cells, and the fluorescence was measured after three days. When diluting the EV solution, the percentage of edited cells (i.e., EGFP-negative cells) also decreased almost linearly, suggesting a concentration-dependent gene editing efficiency (Fig. 2d, e).

Next, to directly compare the efficiency of EV and AAV, we conducted a parallel experiment using AAV carrying Cas9 and *EGFP*-targeting sgRNA (Fig. 2f, g, Supplementary Fig. 3d). Cas9 and *EGFP*-targeting sgRNA encoding AAV vectors were packaged with AAV serotype 2, which has been previously reported to show high transduction efficiency in HEK-293 cells[32]. Upon diluting the *EGFP*-targeting AAV

solution, we observed a reduction in the proportion of EGFP-negative cells three days after transduction, similar to the results obtained with EVs (Fig. 2g, Supplementary Fig. 3d). Notably, the AAV group exhibited a significantly higher variance than the EV group (Supplementary Fig. 3e, p < 0.0001, F-test). These findings suggest that EVs could serve as a viable alternative to AAV for quantitative in vivo genome editing, prompting our exploration of their application in investigating neural function.

## In vivo genome-editing in brain tissue through EV-mediated Cas9/sgRNA RNPs delivery

Next, to examine whether EV-mediated in vivo genome editing can be applied to brain tissue, we injected *EGFP*-targeting EVs into the brains of adult EGFP-expressing transgenic mice and quantified gene editing efficiency. We targeted the NAcS of Drd1-EGFP mice, a bacterial artificial chromosome (BAC)-transgenic mouse line expressing EGFP under the control of the dopamine receptor D1 (Drd1) promoter[33] (Fig. 3a). Approximately 58% of neurons in the NAcS express EGFP in the Drd1-EGFP mice[34]. Red fluorescent signals from Cas9-fused CherryPicker were detected 24 h after surgery but not after 5 weeks, indicating transient expression of the Cas9 protein (Fig. 3b, c). The EGFP fluorescent intensity and the number of EGFP-positive cells of the NAcS injected with *EGFP*-targeting EVs was significantly decreased, compared to those injected with the negative control EVs (Control, 16.5 ± 0.8; *EGFP*-targeting EV, 12.4 ± 1.2) (Fig. 3c, d, Supplementary Fig. 3f–h), indicating successful in vivo genome-editing via EV. In addition, similar to the in vitro experiment (Fig. 2e), the intensity of

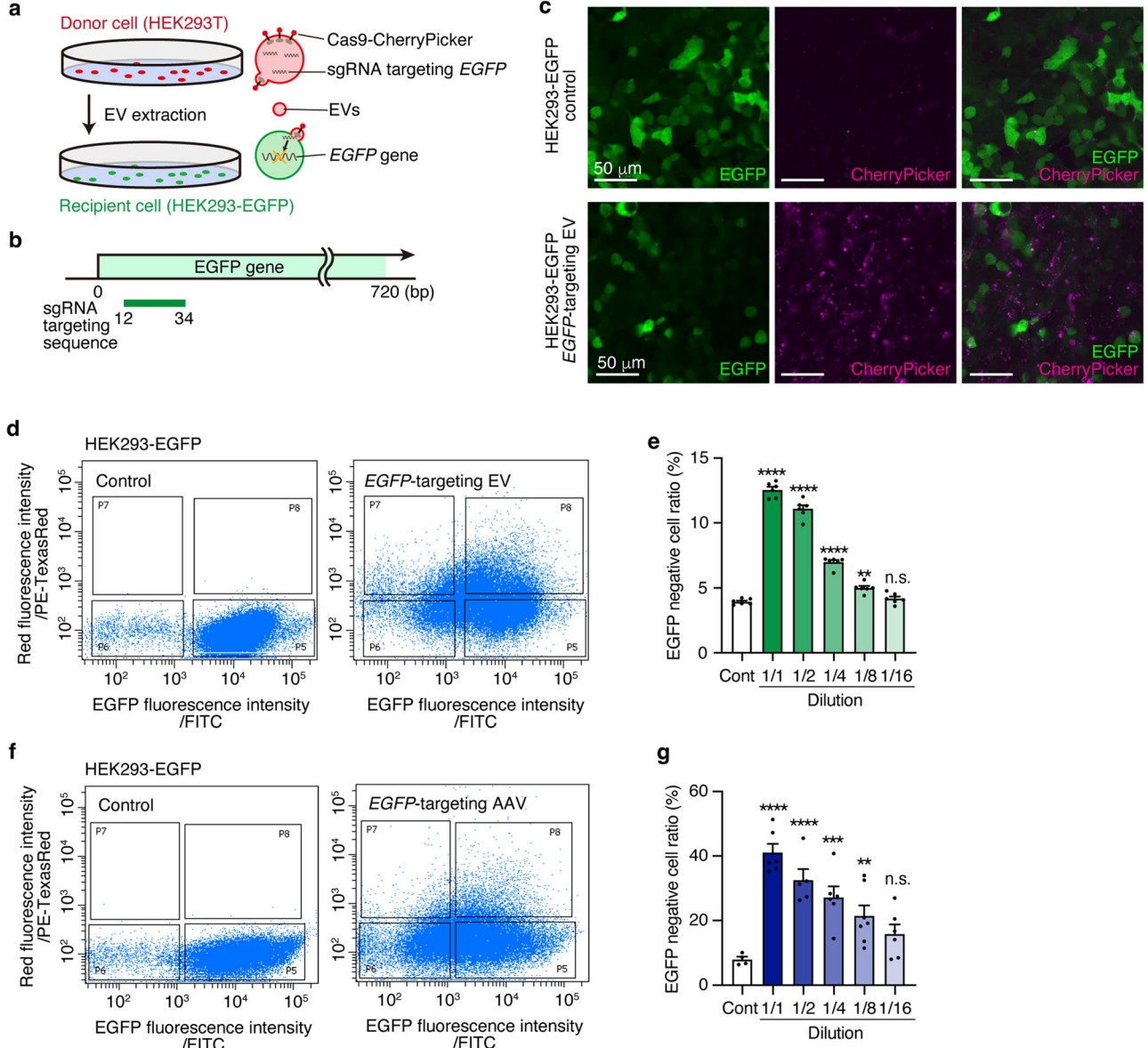

**Fig. 2 | EV delivering Cas9/sgRNA RNPs edited *EGFP* gene of HEK293-EGFP in vitro. a** Schematic illustration of the *EGFP* gene targeting Cas9/sgRNA RNPs containing EV production and transduction. **b**. Schematic illustration of the position of the *EGFP* gene targeting sgRNA on the *EGFP* gene. **c** Representative confocal microscopy images of HEK293-EGFP (upper panel) and *EGFP*-targeting EV (EV-#3) transduced HEK293-EGFP cells. GFP (green) expressed in the HEK293-EGFP cell and Cas9-fused CherryPicker (Magenta) signal were derived from EV. Scale bar = 50 μm. **d** Representative raw flow cytometry data of untreated HEK293-EGFP cells (left panel) and *EGFP*-targeting EV transduced HEK293-EGFP cells (right panel). **e** Quantification of EGFP negative cell ratio of the *EGFP*-targeting EV transduced

HEK293-EGFP cells with varying concentrations (Control, 1/1–1/16: $n = 6$ samples. One way ANOVA followed by Dunnett's multiple comparisons test). **f** Representative raw flow cytometry data of untreated HEK293-EGFP cells (left panel) and *EGFP*-targeting AAV transfected HEK293-EGFP cells (right panel). **g** Quantification of EGFP negative cell ratio of the AAV transfected HEK293-EGFP cells with varying concentrations (Control: $n = 4$ samples, 1/1: $n = 6$ samples, 1/2: $n = 5$ samples, 1/4: $n = 6$ samples, 1/8: $n = 7$ samples, 1/16: $n = 6$ samples. One way ANOVA followed by Dunnett's multiple comparison test). $*p < 0.05$, $**p < 0.01$, $***p < 0.001$, $****p < 0.0001$. n.s. not significant. Data presented as mean ± SEM. Source data are provided as a Source Data file.

EGFP decreased in a concentration-dependent manner (Fig. 3e), suggesting the possible use of the system to control the proportion of the gene-edited cell population by diluting EV.

### Concentration-dependent EV-mediated cKO of *Shank3* in the vCA1

Next, we generated EVs containing Cas9/sgRNA RNPs targeting *Shank3* (*Shank3*-cKO-EV) to quantitatively examine the pivotal functions of Shank3 in social memory. An sgRNA targeting sequence that encodes the PDZ domain, which is a functionally essential domain comprising all main Shank3 isoforms (Shank3a–d) and is known to interact with

other synaptic proteins[35], was chosen. *Shank3*-cKO-EV was bilaterally injected into the hippocampal vCA1 region to target the pyramidal cell layer of vCA1 (Fig. 4a, b), followed by dissection of the ventral part of the hippocampus (vHPC) containing vCA1 5 weeks after injection. We confirmed various mutations, including in-frame missense and nonsense mutations, induced in the target sequence of *Shank3* by Sanger sequencing (Supplementary Fig. 4a, b) and verified the efficiency of *Shank3*-cKO-EV using quantitative real-time PCR (qRT-PCR) (Fig. 4d). We stained the Shank3 protein by immunohistochemistry using antibodies targeting synaptic Shank3[36] to further validate *Shank3* knockout efficiency at the protein level (Supplementary Fig. 4c–e). Shank3

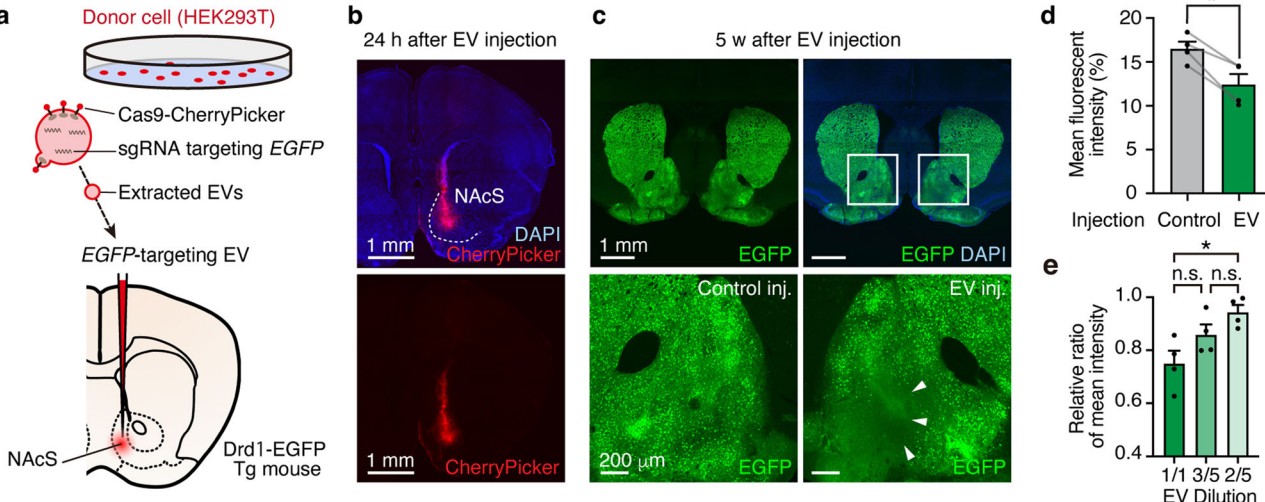

**Fig. 3 | *EGFP*-targeting EVs edit *EGFP* gene in NAcS of Drd1-EGFP mice in vivo.** **a** Schematic illustration of the *EGFP* gene targeting EV injection in NAcS of Drd1-EGFP mice. **b** Representative confocal microscopy image of EV injected NAcS 24 hours after injection stained with anti-RFP (red, for Cas9 fused CherryPicker expression) and DAPI (blue). Scale bar = 1 mm. **c.** Representative confocal microscopy image of EV injected NAcS of a Drd1-EGFP mouse 5 weeks after injection, stained with anti-GFP (green, for EGFP) and DAPI. Macro view of EV-injected NAcS (upper panels), the EV-injected side (lower right panel), and the control injected side (lower left panel). Scale bar = 1 mm (upper panels); 200 μm (lower panels). **d**. Quantification of mean fluorescent intensity of control injected side and EV injected side (n = 4 samples. Paired t-test, two-tailed). **e** Quantification of relative ratio of mean intensity (EV injected site / Control site) in varied EV concentration (1/1–2/5: n = 4 samples. One way ANOVA followed by Tukey's multiple comparison). *$p < 0.05$, **$p < 0.01$, ***$p < 0.001$, ****$p < 0.0001$. n.s. not significant. Data presented as mean ± SEM. Source data are provided as a Source Data file.

puncta were detected in the dendritic spines of social memory engram neurons (Supplementary Fig. 4e). The knockout efficiency of *Shank3* could be quantitatively controlled by diluting the EV (Fig. 4e, f), which was consistent with the validation experiment with *EGFP*-targeting EV (Fig. 3d, e).

To investigate the presence of a threshold proportion of cells in which *Shank3* is edited to cause social memory impairment, mice with different percentages of *Shank3* knockout in the vCA1 were generated. Based on the results shown in Figs. 2e, 3e, and 4f, the knockout efficiency of the target gene could be quantitatively controlled by altering the concentration of the injected EV. Thus, we serially diluted the *Shank3*-cKO-EV solution (dilution factors: 1/2, 1/4, 1/8, 1/16, 1/32, 1/64, 1/128, 1/256, and 1/512), bilaterally injected each solution, as well as non-diluted *Shank3*-cKO-EV and control-EV into vCA1, and performed a social discrimination test to assess social memory. The control-EV-injected subject mice showed a significantly longer interaction time with novel individuals than with familiar individuals, whereas the non-diluted *Shank3*-cKO-EV injected subject mice did not show a preference for novel individuals (i.e., social memory impairment), similar to the previous experiments with AAV-mediated *Shank3*-cKO mice (Fig. 1g, h). Moreover, the social memory score of the diluted EV-injected group suggested the recovery of memory-dependent social discriminatory behavior by increasing the dilution factor of *Shank3*-cKO-EV (Fig. 4g, h). Specifically, the non-diluted (1/1) group demonstrated no preference for novel mice compared with the 1/64 group, whereas the 1/128, 1/256, and 1/512 groups showed normal social discrimination behavior. Together, these results suggested that the threshold of *Shank3*-unedited neurons for obtaining normal social memory exists between 1/64 and 1/128 (Fig. 4i).

## Discussion

Here, we used the AAV- and EV-mediated CRISPR/Cas9 systems to demonstrate that the specific cKO of *Shank3* in hippocampal vCA1 neurons disrupts social memory. It is worth noting that *Shank3*-cKO in vCA1 is sufficient to recapitulate the impairments in social discriminatory behavior observed in conventional *Shank3* knockout mice[12,25]. vCA1[18,37], upstream dCA2[19,38], and hippocampal microcircuits

between these regions[24] all play indispensable roles in social memory[17]. In addition, afferent inputs to the dCA2[20,22] and efferent projections from the vCA1[18,39] modulate social memory and social discriminatory behavior. Several studies have implied a relationship between these hippocampal regions and impaired social memory in individuals with ASD and other related disorders. A series of studies using a mouse model of 22q11.2 deletion syndrome, a chromosomal risk factor for multiple neuropsychiatric disorders, including autism-associated behavioral features, demonstrated impaired social memory and disrupted social novelty coding in the CA2 region[40,41]. In contrast, hyperactivity of projections from the vCA1 to the medial prefrontal cortex (mPFC) in autism-associated Mecp2 knockout mice, which show social memory deficits, has also been reported[39]. Together with our previously published paper[25], our data support the idea that vCA1 is a pathophysiological candidate region for social memory impairment in neurodevelopmental disorders, including ASD.

More importantly, our results suggest that a deficiency in *Shank3* expression above a critical proportion of vCA1 neurons leads to social memory impairment. In other words, a certain number of unedited (i.e., normal) neurons are required to obtain social memory and exhibit typical social discriminatory behavior. We demonstrated the possible existence of a threshold by varying the concentration of Cas9/sgRNA RNP-containing EV. In recent years, the utility of EVs as a natural cargo for drug delivery has been extensively studied[42]. Parallel to technological developments, EV-mediated Cas9/sgRNA RNPs delivery for the transient application of the CRISPR/Cas9 system to cells has also been reported[30,43–46]. In the present study, we used an EV system for in vivo genome-editing[30]. Our study shows the in vivo application of EV to brain tissue for genome editing and proposes EV as a tool to evaluate gene function in a specific brain region in a gradient manner.

EV-mediated delivery of Cas9/sgRNA RNPs into the brain tissue has several benefits that may overcome the methodological limitations of conventional AAV-mediated genome-editing systems. One is from features of the transiency and rapidity of the system. The EV-mediated system directly carries the Cas9 and the sgRNA as protein and RNA, respectively, whereas the AAV-mediated delivery system involves DNA vectors encoding Cas9 and sgRNA, which are later transcribed and

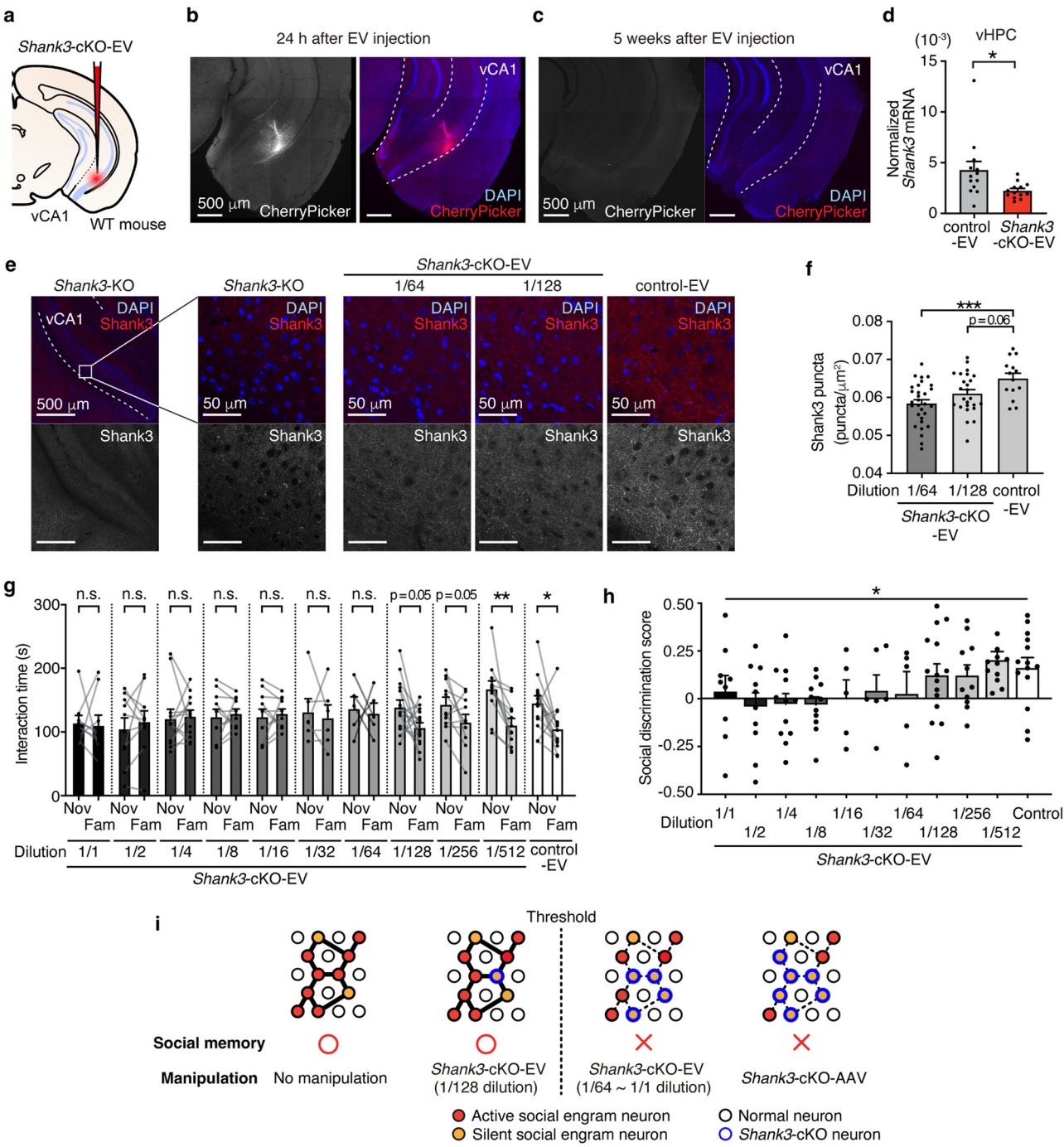

**Fig. 4 | A threshold of *Shank3*-unedited neurons to obtain normal social memory exists. a** Schematic illustration of the injection of *Shank3*-cKO-EV into vCA1. **b-c.** Representative confocal microscopy image of a *Shank3*-cKO-EV injected vCA1 24 h following injection (**b**) and 5 weeks after injection (**c**). Scale bar = 500 μm. **d** *Shank3* mRNA normalized with beta-actin (*Actb*) mRNA quantified by qRT-PCR (control-EV: *n* = 13 samples; *Shank3*-cKO-EV: *n* = 13 samples. Unpaired t-test, two-tailed. see Methods). **e** Representative confocal microscopy images of vCA1 stained with anti-Shank3 (red) and DAPI (blue). *Shank3*-KO mouse, *Shank3*-cKO EV injected mouse (1/64 and 1/128), and control-EV injected mouse, respectively **f.** The number of Shank3 puncta in vCA1 of *Shank3*-cKO-EV injected mice (1/64: *n* = 30 sections; 1/128: *n* = 25 sections; Control: *n* = 14 sections. One-way ANOVA followed by

Dunnett's multiple comparisons test). **g** Social discrimination test of *Shank3*-cKO-EV injected mice (1/1: *n* = 9 mice; 1/2: *n* = 10 mice; 1/4: *n* = 12 mice; 1/8: *n* = 11 mice; 1/16: *n* = 5 mice; 1/32: *n* = 6 mice; 1/64: *n* = 5 mice; 1/128: *n* = 15 mice; 1/256: *n* = 11 mice; 1/512: *n* = 12 mice; Control: *n* = 14 mice. Two-way mixed model ANOVA). **h** Social discrimination score of *Shank3*-cKO-EV injected mice (1/1: *n* = 9 mice; 1/2: *n* = 10 mice; 1/4: *n* = 12 mice; 1/8: *n* = 11 mice; 1/16: *n* = 5 mice; 1/32: n = 6 mice; 1/64: *n* = 5 mice; 1/128: *n* = 15 mice; 1/256: *n* = 11 mice; 1/512: *n* = 12 mice; Control: *n* = 14 mice. One-way mixed model ANOVA). **i.** Schematic illustration of a working model of the required cell population for obtaining social memory in vCA1. *$p < 0.05$, **$p < 0.01$, ***$p < 0.001$, ****$p < 0.0001$. n.s. not significant. Data presented as mean ± SEM. Source data are provided as a Source Data file.

translated inside the target cell. It is known that approximately 0.1 % of vectors enter and integrate via non-homologous recombination in the host genome[47,48]. In contrast, the EV-mediated genome-editing method is free from undesired genome integration and its consequences

because the system delivers the functional RNPs themselves, not the DNA vectors[30,49]. Moreover, whereas conventional single-stranded AAV vectors require several days to produce functional proteins after infection, an EV-mediated system can immediately and directly deliver

functional RNPs into the host cell within 1 h and then degrade 24 h after the application in vitro[30,50,51]. Another benefit is from the features of EV as a lipid bilayer vesicle. EVs have relatively fewer safety issues in terms of immunogenic reactions than viral-mediated methods[49]. Furthermore, unlike AAV, EVs are not occasionally transported in a retrograde and/or anterograde direction, which prevents undesirable protein expression in non-target brain regions/tissues.

Despite these benefits, EV-mediated genome-editing systems have several challenges to be used in a wide range of studies. First, the genome-editing efficiency of the EV-mediated system, which enables the transient expression of editing molecules was lower than that of AAV mediated one, which enables prominent and persistent expression. Secondly, targeting specific cell types using EV-mediated RNPs delivery is challenging. AAV-mediated systems can efficiently target specific cell types when combined with genetic techniques (e.g., designed promoters and Cre/lox recombination)[52–54]. In contrast, the EV-mediated system used in our study is reported to target all cell types[30]. Because it is still unclear whether the different subtypes of EVs intrinsically target different cell types or whether the uptake process is generally stochastic and unspecific[55], it is necessary to understand EV-based signal transduction itself in nature, to develop an EV-mediated system to label specific cells. In this study, we presented an alternative option for the in vivo genome-editing method in the brain tissue, which can be useful depending on its purpose.

To date, no prior study has definitively determined the number of *Shank3*-unedited neurons are required to obtain social memory. Our group has previously reported that the optogenetic reactivation of vCA1 social engram neurons encoding a specific social memory is sufficient to artificially retrieve the social memory[18]. In contrast, partial reactivation of the vCA1 social memory neurons (approximately 20% of the labeled neural population) by halving the laser intensity failed to recall social memory. These results suggest that a certain proportion of social memory neurons must be reactivated to retrieve social memories and demonstrate social discriminatory behavior. In this study, we exhibited a certain threshold of *Shank3*-unedited (i.e., *Shank3*-expressing) neurons, regardless of neural projection patterns, to show normal social memory function. Although it was technically difficult to examine the exact percentage of *Shank3*-edited neurons in mice injected with diluted *Shank3*-cKO-EV, we could discuss the presence of the threshold in terms of both optogenetic reactivation and Shank3 function in social engram neurons (Fig. 4i).

Social memory, a component of episodic memory, is thought to follow the general mechanism of memory formation, in which the cellular mechanisms of synaptic potentiation are necessary. Multiple in vitro electrophysiological studies have demonstrated altered synaptic transmission and reduced long-term potentiation in the CA1 region, with no difference in long-term depression (LTD) in *Shank3* mutant mice[56–58]. Shank3 deficiency alters the expression of mGluR5 and changes the AMPA/NMDA ratio[57,59,60]. Although the specific electrophysiological phenotype depends on the type of mutation in *Shank3* in each study, these molecular alterations in postsynaptic density are thought to ultimately change the balance between excitation and inhibition (E/I balance) of neurons, leading to malfunctions in synaptic transmission and potentiation[61,62]. Thus, Shank3 deficiency in the vCA1 neurons likely impaired the synaptic potentiation of social memory neurons and lead to social memory impairment in our experiments. Notably, AAV-mediated *Shank3* cKO in the LH also impaired social discriminatory behavior. These results suggest that multiple anatomically and functionally connected brain regions are required to achieve normal social memory and social discriminatory behavior. Taken together, our findings of social memory impairment in *Shank3*-cKO mice in the hippocampal vCA1 region and its concentration-dependent deficits in social discriminatory behavior provide evidence to suggest the pathophysiological involvement of vCA1 in the social memory deficits of ASD

and a possible threshold of the proportion of *Shank3*-unedited neurons in obtaining normal social memory.

## Methods

### Animals
All procedures followed the protocols approved by the Institutional Animal Care and Use Committee of the Institute for Quantitative Biosciences, the University of Tokyo. C57BL/6 J, C3H/HeJ, and BALB/c mice were obtained from the Central Laboratories for Experimental Animals, Japan. Drd1-EGFP (Tg(Drd1-EGFP)X60Gsat/Mmmh, RRID: MMRRC_000297-MU), originally generated by Nathaniel Heintz, Ph.D., Rockefeller University, GENSAT, was obtained from the Mutant Mouse Resource and Research Center (MMRRC)[33]. *Shank3*-KO mice (B6.129-Shank3[tm2Gfng]/J, RRID: IMSR_JAX:017688) were obtained from The Jackson Laboratory[12]. All mice were housed under a 12-h (7 am–7 pm) light/dark cycle, $23 \pm 2$ °C, 50 % humidity, with *ad libitum* access to food and water. All the wild-type (WT) mice described in this study were C57BL/6 J mice. C3H/HeJ and BALB/c mice were used as demonstrators. Adult male WT mice (3–5 months old) were used as subjects for behavioral assays. Juvenile male C3H/HeJ and BALB/c mice (5–8 weeks old) were used as demonstrators. Adult male Drd1-EGFP mice (3–5 months old) were used for the in vivo confirmation of EV-mediated Cas9/sgRNA RNP delivery. Adult male *Shank3*-KO (3–5 months old) mice were used for Shank3 antibody confirmation. For the social memory engram neuron labeling experiment, male C57BL/6 J mice (7 weeks old) were fed with food containing 40 mg/kg doxycycline (Dox) for one week prior to surgery. The mice continued to be on Dox for the duration of the experiment, except on the neuron labeling day. All mice were sexually and experimentally naïve.

### Genotyping
Subject mice were anesthetized with isoflurane inhalation, and approximately ~5 mm of tail tip tissue was collected for genomic DNA extraction[63]. The supernatant from the digested mixture was used as the PCR template. Quick Taq HS DyeMix (Toyobo) was used for DNA fragment amplification (see Supplementary Table S2 for primer sequences), and products were analyzed by standard gel and capillary electrophoresis (MultiNA, Shimadzu).

### Social discrimination test
The social discrimination test (SDT) protocol was modified from the original SDT[18]. Briefly, 10–16 weeks old (25–30 g weighted) WT and *Shank3*-KO male mice were used for behavioral experiments. The subject mice were individually habituated to the experimenter over three days. Habituation to the open field test chamber (380 mm × 380 mm × 300 mm) with two mouse holders (a quarter cylinder shape with a radius of 7 cm and height of 10 cm) on opposite corners was performed for 10 min each day. C3H/HeJ and BALB/c (5–8 weeks old; 20–25 g weighted) male mice were used for demonstrators. These mice were pre-handled for at least three days before the experiment. On the afternoon of the last day of habituation, a demonstrator mouse (either C3H/HeJ or BALB/c) was placed into the home cage of the subject mouse for familiarization (total duration 72 h). On the experimental day, a familiarized demonstrator mouse was separated for 30 min before behavioral recording. Familiar and novel demonstrator mice of a different strain (e.g., C3H/HeJ as a familiar demonstrator mouse, BALB/c as a novel demonstrator mouse and vice versa) were placed inside each mouse holder. A subject mouse was then placed in the chamber and tracked for 10 min using EthoVision XT (Noldus). The chamber was cleaned using ion-filtered water between sessions, and each experiment was counterbalanced. The social discrimination score was calculated using the following formula: $Social\ discrimination\ score = \frac{Interaction\ time\ (Novel-Familiar)}{Interaction\ time\ (Novel+Familiar)}$. All the behavior experiments were performed in 3–5 batches and then combined.

## Surgery

Mice were anesthetized using M/M/B mixed anesthetics (0.75 mg/kg medetomidine, 4.0 mg/kg midazolam, and 5.0 mg/kg butorphanol) and mounted on a stereotaxic apparatus (Leica Angle Two, Leica Biosystems). The viral and EV solutions were microinjected using a glass pipette attached to a microsyringe (Hamilton) filled with mineral oil. A microsyringe pump injector (UMP3, World Precision Instruments) and its controller were used to control the speed and the amount of solution delivered. The glass pipette was slowly lowered to the target site stereotaxically, and the solutions were delivered at a speed of 2-3 nl/sec and retracted 5 minutes after injection.

For AAV- and EV-mediated genome-editing experiments, bilateral solution delivery was aimed at coordinate relatives to Bregma: vCA1 injections were targeted to −3.16 mm Anterior-Posterior (AP), ±3.10 mm Medial-Lateral (ML), and −4.70 mm Dorsal-Ventral (DV); vmPFC injections were targeted to +1.80 mm AP, ±0.35 mm ML, and −2.50 mm DV; NAc injections were targeted to +1.34 mm AP, ±0.60 mm ML, and −4.60 mm DV; LH injections were targeted to +1.80 mm AP, ±0.65 mm ML, and −5.65 mm DV. For AAV-mediated conditional knockout experiments, 300 nl of AAV9-CMV:Cas9-T2A-mCherry/U6:sgRNA (control or *Shank3*-targeting, $2 \times 10^{12}$ genome copies [GC]/ml) was injected into each region. For the projection-dependent *Shank3* knockout experiment, 85 nl of AAV1-hSyn:Cre (Addgene, $2 \times 10^{13}$ GC/ml) was injected into vCA1 and 200 nl of a 1:1 cocktail of AAV9-pMeCP2:DIO-Cas9 ($2 \times 10^{12}$ GC/ml) and AAV9-U6:sgRNA(*Shank3*)/hSyn:DIO-EYFP ($2 \times 10^{12}$ GC/ml) (cKO) or AAV9-U6:sgRNA(*Shank3*)/hSyn:DIO-EYFP ($2 \times 10^{12}$ GC/ml) (Control) was injected into the LH. For in vivo *EGFP*-targeting EV-mediated genome-editing experiments, *EGFP*-targeting EV-#3 was injected into the NAc (+1.34 mm AP, ±0.60 mm ML, and −4.60 mm DV). For the in vivo EV-mediated *Shank3*-cKO experiments, 500 nl of non-diluted and diluted *Shank3*-targeting EV was injected into bilateral vCA1. For the social memory engram neuron labeling experiment, 300 nl of 1:1 cocktail of AAV9-c-fos:tTA ($2.0 \times 10^{13}$ GC/ml) and AAV9-TRE:ChR2-EYFP ($2.0 \times 10^{13}$ GC/ml) was injected into bilateral vCA1. After the surgeries, 0.5 mg/kg carprofen and 0.75 mg/kg atipamezole were intraperitoneally injected into the mice. Brain illustrations are based on images from *"Paxinos and Franklin's the Mouse Brain in Stereotaxic Coordinates"*.

## Histology and immunohistochemistry

Mice were first anesthetized using M/M/B mixed anesthetics. After confirming proper anesthesia, the mice were transcardially perfused with 4% paraformaldehyde (PFA) in phosphate-buffered saline (PBS). Brains were post-fixed overnight in 4% PFA in PBS, sectioned to 50 μm using a vibratome (VT1000S, Leica), and collected every three sections. Sections were incubated in 0.3% Triton-X in PBS (-) with 5% normal goat serum for 1 h at room temperature, and then incubated with primary antibody overnight at 4 °C. The primary antibodies used were rabbit anti-RFP (600-401-379, 1:1000; Rockland), chicken anti-GFP (A10262, 1:1000; Thermo Fisher Scientific) and rabbit anti-Shank3 (64555, 1:500; Cell Signaling Technology). After washing with PBS (-) once for 5 min followed by thrice for 15 min each, brain sections were incubated with anti-chicken Alexa Fluor-488 (A11039, 1:500; Thermo Fisher Scientific), antirabbit Alexa Fluor 546 (A11010, 1:500; Thermo Fisher Scientific), or anti-rabbit Alexa Fluor 488 (A11008, 1:500; Thermo Fisher Scientific) conjugated secondary antibodies in 0.3% Triton-X in PBS (-) with 5% normal goat serum for 3 h at RT. Sections were washed twice with PBS (-)for 5 and 15 min. All samples were stained with 4′,6-diamidino-2- phenylindole (DAPI) (1 μg/mL) for 15 min, and washed in PBS (-) for 15 min. The sections were then mounted on glass slides using Vectashield (Vector Laboratories). Images were taken using a confocal laser microscope (FV3000, Olympus) with 10X and 20X objectives and by fluorescence microscopy (BZ-X710, Keyence) with a 10X objective. Images were post-processed using ImageJ (NIH).

## Cell maintenance and harvesting

HEK-293T (Takara cat. no. 632617), HEK293-EGFP (GenTarget, Cat. No. #SC001), and NIH-3T3 (originally obtained from Riken Bioresource Research Center and gifted from Dr. Atsushi Miyajima, University of Tokyo) cell lines were grown in high glucose DMEM (10% FBS and 1% penicillin/streptomycin supplemented) at 37 °C and 5% $CO_2$.

## Cas9 plasmid construction and AAV production

The pAAV-CMV:Cas9-T2A-mCherry/U6:sgRNA plasmid was modified from pX601 (pAAV-CMV:NLS-SaCas9-NLS-3xHA-bGHpA;U6::BsaI-sgRNA; Addgene plasmid # 61591). To visualize the AAV-infected cells, the T2A-mCherry sequence was amplified by PCR and added after the second NLS sequence using the NEBuilder HiFi Assembly Master Mix (New England Biolabs). An sgRNA sequence targeting exon 2 of *Shank3* (Supplementary Table 2, Shank3-sgRNA-1)[13] was annealed and was packaged in-house with AAV serotype 5.

The pAAV-pMeCP2:DIO-Cas9 plasmid was modified from pX551 (pAAV-pMeCP2:SpCas9-spA; Addgene plasmid # 60957) and was gifted by Dr. Dong Kong[64]. The pAAV-U6:sgRNA/hSyn:mCherry and the pAAV-U6:sgRNA/hSyn:DIO-EYFP plasmids were modified from pX552 (pAAV-U6:sgRNA(SapI)/hSyn-GFP-KASH-bGH; Addgene plasmid # 60958). To make this modification, the original plasmid was digested using BamHI and EcoRI restriction enzymes. The PCR amplified mCherry sequence or the (BamHI)-DIO-EYFP-(EcoRI) fragment, which is a restriction enzyme digested product of pAAV-Ef1α:DIO-EYFP (Addgene plasmid # 27056), were then ligated using T4 DNA ligase (NEB) according to the standard protocol. *Shank3*-targeting sgRNA (Supplementary Table 2, Shank3-sgRNA-2) was annealed, and constructs were packaged in-house with AAV serotype 9. For the AAV transduction experiment into HEK293-EGFP cells, the pAAV-U6:sgRNA/hSyn:mCherry was modified from pX552. *EGFP*-targeting sgRNA (Supplementary Table 2, EGFP-sgRNA-3) sequence was annealed. DNA constructs, together with pX551, were packaged in-house with AAV serotype 2.

The pAAV-c-fos:tTA and pAAV-TRE:ChR2-EYFP plasmids[65], an AAV based activity-dependent cell labeling method, were used for the social memory engram labeling experiment. These plasmids were packaged in-house with AAV serotype 9.

The pAAV plasmids with the AAV helper plasmid (pAdDeltaF6; Addgene plasmid # 112867) and pAAV2/2 (for AAV2; Addgene plasmid # 104963), pAAV2/5 (for AAV5; Addgene plasmid # 104964), or pAAV2/9n (for AAV9; Addgene plasmid # 112865) were co-transfected into HEK-293T cells using PEI-Max for AAV production. AAV was purified 3–5 days after transfection using AAVpro Purification Kit Midi or Maxi (Takara Bio). The viral concentration was measured by qRT-PCR.

## Cas9 delivering extracellular vesicle production

"Gesicle," a specialized EV Cas9/sgRNA complex delivery system, was produced and purified according to the Guide-it CRISPR/Cas9 Gesicle Production System protocol (Takara Bio) involving oligo-DNA pair annealing in a thermal cycler.

The annealed oligos were ligated with the pGuide-it-sgRNA1 Vector and prepared using the FastGene Plasmid Mini Kit (Nippon Genetics) and NucleoBond Xtra Midi (Takara Bio). Transfection into HEK-293T cells was performed according to the manufacturer's instruction. After 72 h of incubation, the supernatant was collected, centrifuged (500 × g, 5 min), and filtered using 0.45 μM sterile filters. Filtered supernatant was centrifuged again at 4 °C, 8000 × g for 18 h. After discarding the supernatant, the pellet was resuspended in 60 μl of PBS (-), incubated at 4°C for 2 h, and aliquoted.

*EGFP*-targeting sgRNA encoding plasmid #1 was created by targeting 32–54 bp of the *EGFP* gene, plasmid #2 targeted 52–74 bp, plasmid #3 targeted 12–34 bp, and plasmid #4 targeted 73–95 bp of the *EGFP* gene (see Supplementary Table S2 for oligo sequences). The

*Shank3*-targeting sgRNA encoding plasmid for *Shank3*-cKO-EV was created targeting 32001–32021 bp of the *Shank3* gene (Supplementary Table S2). Purified *Shank3*-cKO-EV was transduced into NIH-3T3 recipient cells, followed by a T7 endonuclease I (T7EI) assay using the Guide-it Mutation Detection Kit (Takara Bio) following the standard protocol for mismatch detection.

### In vitro EV transduction

HEK293-EGFP and NIH-3T3 cell lines were subcultured in 24-well collagen I-coated plates at a density of 10,000 cells/500 μl per well the day before transduction. Protamine sulfate (Final concentration: 8 μg/ml, Takara Bio) was added, and 30 μl of EVs targeting EGFP or *Shank3* gene were applied to each well. The plates were centrifuged at 1,150 g for 30 min at RT and incubated at 37 °C for at least 3 h initially. Subsequently, the medium was replaced to D-MEM without protamine sulfate, and continued incubation at 37 °C for additional 69 h and prepared for flow cytometry analysis. In order to maintain consistent and precise control of genome-editing efficiency, we used a single batch of EV solution for each experiment.

### In vitro AAV transfection

First, a 1:1 cocktail of AAV2-pMeCP2-Cas9 ($5 \times 10^{12}$ GC/ml) and AAV2-U6:sgRNA(target: *EGFP*)/hSyn:mCherry ($5 \times 10^{12}$ GC/ml) was prepared. AAV2-U6:sgRNA(target: *EGFP*)/hSyn:mCherry was used as the control AAV. Then, 1 μl of non-diluted and diluted AAV solutions were directly applied to HEK293-EGFP cells with 40-50% confluency, grown in 24-well collagen I-coated plates. The cells were incubated overnight at 37 °C, and then the medium containing AAV was replaced with fresh D-MEM. The cells were incubated for a total of 72 h and prepared for flow cytometry analysis.

### Flow cytometry (FACS)

Wells were washed twice with 300 μl of PBS (-) and incubated with 500 μl of trypsin at 37 °C for 3 min. Then, 500 μl of 3% FBS/PBS (-) was added to each well, and plates were centrifuged at 600 g for 8 min at RT in a sterile 1.5 ml tube to pellet the cells. The cell pellets were suspended in 1 ml of 3% FBS/PBS (-), and the suspended solution was filtered through a 35-μm cell strainer (Falcon) to remove aggregated cells. The fluorescence intensity of the filtered samples was measured using a FACS Aria III cell sorter (BD Biosciences). Flow cytometry was performed on EV- or AAV-treated HEK293-EGFP cells for comparing the genome-editing efficiency of sgRNA (#1-#4) and the effects of serial dilution. EGFP fluorescence intensity less than $2 \times 10^3$ (au) was defined as EGFP-negative, and CherryPicker red fluorescence intensity less than $3 \times 10^2$ (au) as CherryPicker-negative cells. The EGFP$^{Positive}$/CherryPicker$^{Negative}$, EGFP$^{Negative}$/CherryPicker$^{Negative}$, EGFP$^{Negative}$/CherryPicker$^{Positive}$, and EGFP$^{Positive}$/CherryPicker$^{Positive}$ cell groups were defined as P5, P6, P7, and P8, respectively. The EGFP-negative cell ratio was calculated using the following equation: $EGFP\ negative\ cell\ ratio\ (\%) = \frac{The\ number\ of\ cells\ (P6 + P7)}{The\ number\ of\ cells\ (P5 + P6 + P7 + P8)}$.

### Labeling of social memory engram neurons

72 hours before the social memory labeling experiment, the food containing Dox was replaced with normal food without Dox (OFF Dox). Age-matched, male WT (demonstrator) mice were then introduced into the home cage of the subject mice for a 2-hour social interaction. After the demonstrator mice were removed from the subject mice's home cage, the diet was switched back to food containing Dox (ON Dox). The subject mice were perfused 48 hours post-labeling.

### Microscopy analysis

For in vivo EV-mediated genome-editing experiments (related to Fig. 3 and Supplementary Fig. 3), two slides of NAcS were collected from unilaterally EV-injected Drd1-EGFP mice. The fluorescence intensity values (0-255) were obtained using ImageJ (NIH) by measuring the mean intensity of the selected area on 8-bit-converted microscopy images. The values were then divided by the maximum intensity value (255) and expressed as a percentage. The relative ratio of the mean intensity was calculated by dividing the mean intensity (%) with that of the ipsilateral control side.

For the cell number analysis, microscopy images of *EGFP*-targeting EV-injected NAcS and control were cropped to 500 μm$^2$ using ImageJ. DAPI- and EGFP-positive cells were counted using the StarDist 2D plugin. The size and mean fluorescence intensity of the EGFP-positive region of interest (ROI) were measured on an 8-bit-converted EGFP fluorescence channel for the real dataset and on a different sample for the pseudo-dataset. Only cells within the size range of the mean ± 1 SD and mean fluorescence above the threshold (67.614 out of a maximum of 255; the intersection point of histograms derived from real and pseudo datasets) were included for adjusted cell number.

For Shank3 puncta analysis, three to four sections from each of three individuals were used for immunohistochemistry (Fig. 4, Supplementary Fig. 1, and Supplementary Fig. 4). Areas including vCA1 pyramidal neurons were cropped, and particles from the Shank3 fluorescence channel are analyzed and counted using ImageJ function "Analyze Particles" after setting a threshold. The number of particles was divided by the size of area.

### mRNA extraction and quantitative RT-PCR (qRT-PCR)

Subject mice were deeply anesthetized using isoflurane inhalation (3–4%) and then immediately decapitated for whole brain extraction. Bilateral hippocampi were dissected and divided into three sections, and the ventral part of the hippocampus was used as the vHPC. Tissues were stored at −80°C and homogenized on dry ice, and TRIZOL reagent was used to extract mRNA. The SuperScript III First-Strand Synthesis System (Thermo Fisher Scientific) was used following the standard protocol with polyA primers for synthesizing cDNA. qRT-PCR was performed using the Thunderbird SYBR qPCR Mix (Toyobo) and Light Cycler 480 (Roche) following the standard protocol for absolute quantification (see Supplementary Table S2 for primer sequences). The quantified amount of *Shank3* mRNA was divided by the quantified amount of beta-actin (*Actb*) mRNA from the same sample for normalization[66].

### Statistical analysis

Statistical analyses were performed using Prism 9 (GraphPad) and MATLAB (MathWorks).

### Statistics and reproducibility

Suitable sample sizes were determined based on the Cohen's d effect size and our previous study as well as other's similar studies which are generally employed in the field of study[18,19]. All behavioral experiments were conducted in at least three different batches and all batches showed a similar trend. All the cell culture experiments were conducted in at least two different batches and all batches showed a similar trend. All fluorescent image analyses were independently repeated at least twice and consistently demonstrated a similar trend.

### Reporting summary

Further information on research design is available in the Nature Portfolio Reporting Summary linked to this article.

## Data availability

Data supporting the findings in the present study are available from the corresponding author upon request. The data generated in this study are provided in the Source Data file. Source data are provided with this paper.

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

## Acknowledgements

We thank Dr. A. Miyajima and Dr. Y. Hori (The University of Tokyo) for providing the NIH-3T3 cell line, C. Koga (The University of Tokyo), Dr. N. Horikoshi and Dr. L. Negishi (The University of Tokyo) from the Kurumizaka laboratory for technical assistance related to flow cytometry experiments. We thank I. Yoshimura, M. Sako, M. T. Tang, A. Watarai, A. Matsuyama and M. Watanabe for technical assistance and all members of the Okuyama laboratory for discussions and support. This work was supported by JST, PRESTO Grant Number JPMJPR1781, JPMJFR2143, and JPMJCR23B1 (to T.O.), JSPS KAKENHI Grant Numbers JP18H02544, JP20K21459, JP21H02593, and JP21H05140 (to T.O.), AMED under Grant Number JP21wm0525018 (to T.O.), Grant-in-Aid for a Japan Society for Promotion of Science (JSPS) postdoctoral PD fellowship (to W.A. and M.W.), Grant-in-Aid for a Japan Society for Promotion of Science (JSPS) doctoral DC fellowship (to M.C. and Z.H.), the Naito Foundation (to T.O.) and SECOM Science and Technology Foundation (to T.O.).

## Author contributions

M.C., K.I. and T.O. designed the research; M.C. and K.I. collected and analyzed the data; T.A. and G.F. developed the AAV-mediated in vivo genome-editing technology. Z.H., A.W., M.W., K.T. and H.E developed the experimental resources; and M.C. and T.O. wrote the paper. T.O. supervised the project.

## Competing interests

The authors declare that this study was conducted in the absence of any commercial or financial relationships that could be construed as a potential conflict of interest.
