## [Peer Review File · Nature Communications]

Conditional knockout of Shank3 in the ventral CA1 by quantitative in vivo genome-editing impairs social memory in miceREVIEWER COMMENTS

Reviewer #1 (Remarks to the Author):

This study reports that Shank3 expression in the ventral CA1 (vCA1) region is important for social memory and that Shank3 editing can lead to social amnesia in a dosage-dependent manner. In addition, this study develops an extracellular vesicle (EV)-based delivery of Cas9/sgRNA RNPs for conditional Shank3 KO (cKO) in the vCA1. The authors also test if the effects of EV1-dependent Shank3 cKO can quantitatively elicit the phenotypes.

In light of the increasing demand for genome editing-based rescue in ASD, this study sets an important stage for future related studies. The experimental design and the quality of the data are solid, and the manuscript was well-written. In addition, this study has substantial clinical implications.

Major comments:

1. The authors need to provide information on the nature of the Shank3 mutation used and on how the mutation leads to impairments in the protein level/structure/function and synaptic structure/transmission/plasticity. It seems that the GGT to TTT mutation converts Gly to Phe in the PDZ domain of Shank3. Is this mutation from a patient? If not, what is the rationale for selecting this particular mutation over other mutations? How does it affect the structure and function/protein-interaction/stability of the PDZ domain; it would be easy to clarify this issue or perform simple modeling/experiments given the known crystal structure/PPIs of the Shank3 PDZ domain. Which splice variants of the Shank3 proteins are selectively affected?
2. The study could also be strengthened by additional molecular/cell biological and electrophysiological data. Is the mutant Shank3 protein properly targeted to excitatory synapses, given that the Shank3 PDZ domain interacts with DLGAP/SAPAP proteins for synaptic localization? Does the decreased Shank3 punta density in the vCA1 region accompany a decrease in the total or synaptic levels of Shank3, which can be tested by immunoblot analysis using Shank1/2/3 antibodies? What happens to the excitatory synaptic transmission (i.e., mEPSCs) in the Shank3-mutated CherryPicker-positive vCA1 neurons?
3. The authors carefully compare different dosages of Shank3-EVs and demonstrate that there are threshold levels for the EVs. The amounts of total experiments that the authors performed are admirable, although not for Figure 4D. However, the vCA1 regions are rather large, and it is unclear whether the authors have targeted a large vCA1 region, or a small and particular vCA1 region, such as those shown in Figure 4, considering the unique projection patterns of the ipsilateral/contralateral CA2-CA1 circuits. These comments also apply to the results in Figure 1.

Minor comments:

1. It is unclear what the authors mean by "actb normalized shank3 mRNA" in Figure 4D legend.

Reviewer #2 (Remarks to the Author):

In this manuscript entitled "Conditional knockout of shank3 in the ventral CA1 by quantitative in vivo genome-editing impairs social memory," the authors apply an in vitro developed vesicle gene editing system in a region specific manner to identify Shank3 expression in the vCA1 neurons as necessary for social memory. The results are novel and interesting in that they show that glutamatergic neurons in the vCA1 are a major contributor to social memory via synaptic mechanisms. The manuscript can be improved by addressing the following concerns:

Major:

1. Figure 1: it is important to show that Shank3 protein levels are reduced in this figure using immunostaining.
2. Figure 1E: can the authors provide some validation for the injection areas for vmPFC, NAcS, and LH as these are also included in the figure?
3. Figure 1G: vCA1 interaction time is reduced for both novel and familiar mice, with the same amount of time as the familiar in the control group; meanwhile LH shows more interaction time similar to the amount of time spent with the novel animal in control. This suggests that LH region knock down more powerfully captures the social memory deficit and the mouse spends more time exploring both animals as novel. Can the authors clarify how to best interpret this discrepancy?
4. Figure 1: The authors claim that Shank3 expression in vCA1 "and it's downstream regions" (line 100) are necessary for social memory behavior, but an effect is only shown with Shank3-CKO injection in the LH, and there are no studies showing the specific neurons that go from vCA1 to LH, as opposed to LH neurons receiving inputs from other brain regions, are responsible for the behavior. In order to make this claim, the authors would need to show a projection specific experiment.
5. Figure 2: The authors suggest that a certain number of neurons would be required to support a social memory engram (line 102), however based on the above findings, one would expect that Shank3 expression in vCA1 > LH neurons are most important for this function. Can the authors clarify why they think the number of neurons, as opposed to type of neurons, or where the neurons project, is a relevant factor for vCA1 social memory function?
6. The authors claim that "AAV-mediated in vivo genome-editing may not be adequate to allow expression of the delivered gene in a particular cell proportion because of the difficulty in controlling infection and expression" (line 109), but do not show how dilution of EV compares to dilution of AAV. A comparison to AAV would justify the use/need of this technology better.
7. Figure 3D,E: The authors specifying that 58% of cells in the NAcS express EGFP, but the data shown are expressed as Mean fluorescent intensity (Figure 3D) and relative ratio of mean intensity (Figure 3E). To make a conclusion about number of cells affected by knock down, a different measure would need to be used. Otherwise the conclusion statement should reflect this read out.
8. Figure 4G: statistics performed are a series of paired t-test, but because the experiment involves two variables, 2 way ANOVA is required to determine statistical significance on dilution on behavior.
9. Figure 4H: Requires a one way ANOVA to say the dilution had an effect on behavior (similar to Figure 1H).
10. Discussion: Line 187-188: authors state that that a critical number of neurons are required to support social memory impairment, but Figure 4 does not provide any quantification of number of neurons affected.

Minor:

Rather than going back and forth between "shank3" and "Shank3", it may better to use "Shank3" referring to the murine gene throughout the manuscript.

Line 38: word missing after "The Shank3"

Line 110: please clarify what is meant by "genomic scar"

Line 134: "the adult brains of EGFP-expressing transgenic mice", should be brains of adult EGFP-expressing transgenic mice"

Line 213: "is complicated to" does not make sense in this sentence

Reviewer #3 (Remarks to the Author):

This manuscript describes experiments in which conditional knockout of shank3 in the ventral CA1 region of the hippocampus was performed using quantitative in vivo gene editing. Effects on social memory were demonstrated. The manuscript is generally well written, however there are some major limitations. Comments below suggest ways in which the manuscript could be improved and extended.

General comments:

1. A direct comparison between transfection efficiencies achieved by AAV and EV methods would be desirable, since the authors state the superiority of the EV method.
2. Dosage-dependency of social function on Shank3-KO EV dosage seems overstated by the authors. The data seems to show instead that there is a threshold – effects appear quite similar across undiluted EV dose to 1/64, only then is there a recovery of social function. The authors do conclude this according to Figure 4I; but the emphasis on dosage dependency prior to this does not seem to be supported entirely by the data.
3. Higher quality microscopy images (especially at higher magnifications) would be good in Figures 3 and 4.
4. N numbers need to be explicitly listed in the manuscript. This omission, together with the fact that n numbers within the same experiment appear to vary widely between groups, give the impression that experiments may have been conducted by expanding certain groups without prior planning. The authors need to indicate all the n values for all groups in each graph/figure. Some n values look underpowered for mouse behavioral tests. Were power calculations performed before the experiments were started?
5. The choice of statistical tests appears sometimes to be suboptimal, or incorrect. Why perform multiple t tests when there is the possibility to perform ANOVAs? Particularly relevant in the social interaction/recognition test.
6. The inconsistency between quantification methods chosen at different points (fluorescence intensity vs. shank3 puncta vs shank3 qRT-PCR) raises questions. The manuscript would benefit from being more consistent across the different parts.
7. Some important data in Figure 4 regarding the different dilutions is missing (see below for details).

Results:

8. In Figure 1, this result is interesting (downstream targets), but seems like it is not explored further by the authors. An analogous experiment using EVs would be interesting, as it points towards wider circuitry involved beyond the ventral hippocampus. In Figure 1G, the authors use a paired t-test, which is inappropriate. Why not RM ANOVA? This would also give information on whether there is a statistical difference between the different injection conditions, not just whether their interaction levels with novel vs. familiar mouse are similar or different. E.g. NAcS injection seems to lead to lower differences between novel/familiar interactions.
9. In Figure 1 the authors should show the injection sites for every region.
10. In Figure 1H, social discrimination score: As above, MWU test despite there being several groups? Why not ANOVA or non-parametric equivalent test (Friedman test)?
11. In Figure 2E: Why are the individual datapoints not shown in this graph (despite being shown in 2F)? And "After 69 hours" – why was this timepoint chosen? Were other timepoints checked? Was a threshold reached at this point?
12. Figure 2E, n=1. The authors mentioned they repeated each experiment three times, but they need to be more clear whether it was n=1, repeated three times (technical replicates)?
13. In Figure 2F: EV concentration: 100%, 50%, 25%, 12.5%, 128 and 6.25% - what concentration/amount was 100%? How was this achieved and quantified? If this 100% is the same as

the one used for EV#3 in 1E; why is the proportion of EGFP negative cells much lower in this experiment (28 % vs. ~13%)? This is a stark difference and should be explained, otherwise it calls into question the consistency of the genome-editing efficiency between different experiments.

14. In Figure 3: Negative control EVs - are these completely empty? Would have been good to also have a marker for these injections, as for the Shank-KO EVs. Was the location of control injections checked in a different way? If not, how can the authors be sure that they are looking/measuring in the exact area (and section) where the injection was at a comparable concentration as the other side with Shank-KO EV?

15. In Figure 3B: Image of Cherrypicker after 5 weeks is missing (text says that there was no Cherrypicker expression after 5 weeks, but there is no image). This should be included. Was DAPI also quantified? High magnification images of DAPI channel should also be shown and cell numbers quantified, otherwise there is the possibility that fluorescence may also be reduced due to dopaminergic cell death after EV injection rather than genome editing. Was only 1 section from 4 animals quantified? OR how was this done? Quantification procedure needs more detailed explanation in methods. Why is fluorescent intensity in %? Normally arbitrary units?

16. In Figure 4D: qPCR shank3: how/to what is this normalised? The controls ~ 4.5; but then methods section mentions "absolute quantification". This needs to be clarified either in Results or Methods section.

17. Furthermore, in Figure 4D the N numbers seem very low, especially control group. The variability in controls is large, suggesting (for example via a power calculation) that larger n values are required.

18. In Figure 4D/4E, why not perform qPCR to quantify Shank3 at the different dilutions as well? In Figure 4E, these images give no information about the dilutions of Shank3-KO used, despite the text implying this. Which dilution are these images from? Where are these images taken in relation to the injection site? The Shank3-KO images seem to show that there is a gradient of expression - is this because the bottom right part is close to the injection site? If so, why was the injection site not focused on? The authors should show a lower magnification image (or drawing if unavailable) of the location.

19. In Figure 4F: Why are the dilutions now referred to as 1/64 and 1/128 - how does this relate to the dilutions used previously and referred to as % values? The significance symbols do not show what is described in the figure legend. The comparison between 1/128 and controls should be denoted as significant, but instead the comparison between 1/128 and 1/64 is annotated with "*". Puncta method of quantifying? Why switching between different methods of fluorescence quantification? Also, what does one data point here represent? One individual? Or one section? In addition, it seems to me the most interesting comparison is missing - the diluted EVs vs. the undiluted EVs (or original dose). The authors make statements about the ability to quantitatively control the genome editing, but the data currently does not convincingly support that claim. In fact, it is surprising that a 1/128 dilution is still effective in reducing Shank3 expression vs. controls, and it begs the question as to how much the full dose reduces the expression in this type of quantification. Also, in the next paragraph, 5 further dilutions are used in injections before behavioral experiments. Were these injections also examined for their effect on Shank3 expression? Why are only the lowest ones shown?

20. In Figure 4G, behavior: N numbers seem on the low side for some groups, but then others have far larger samples sizes. Obviously, the statistical power achieved in the different conditions is not comparable and the analyses with higher n numbers are able to detect smaller effect sizes. Additionally, the authors again only test for differences in interaction time between novel and familiar mice in each condition using a paired t test, whereas a repeated-measures ANOVA would be more appropriate, and differences between groups could be evaluated statistically. This result, arguably the main result of this study, is not very convincing in its current state. N numbers should be increased for

most injection conditions to match 1/128 and control groups, and repeated-measures analysis should be performed.

21. In the Methods: N numbers are missing, they should be explicitly stated somewhere, despite many graphs showing individual values, these are not clear enough.

22. Also in Methods: Social recognition test procedure needs clarifications: "demonstrators"/ "stimulators"? What is the role of "stimulators"? They are a different strain, and much younger? Should the mice not be strain and approximately age-matched?

23. In the Methods, it is unclear: familiar mouse is C3H, but then on testing day the familiar mouse is BALB? When is BALB familiarised?

24. Further Methods query: For 72 hours, C3H are placed in home cage – with the new mouse being so much younger, would this not lead to attacks by the resident mouse? How did the body weight compare between demonstrator mice and the experimental cohort? Were mice single housed or group housed throughout experiments?

25. For immunohistochemistry and quantification procedures, the authors need to provide more detail. How many sections per mouse were quantified? Each mouse should still remain the experimental unit, i.e. if several sections are measured, statistics should be carried out on average values per mouse, not on values per section (shank3 puncta experiment).

Point-by-point response to the reviewer's comments

Reviewer #1 (Remarks to the Author):

This study reports that Shank3 expression in the ventral CA1 (vCA1) region is important for social memory and that Shank3 editing can lead to social amnesia in a dosage-dependent manner. In addition, this study develops an extracellular vesicle (EV)-based delivery of Cas9/sgRNA RNPs for conditional Shank3 KO (cKO) in the vCA1. The authors also test if the effects of EV1-dependent Shank3 cKO can quantitatively elicit the phenotypes.

In light of the increasing demand for genome editing-based rescue in ASD, this study sets an important stage for future related studies. The experimental design and the quality of the data are solid, and the manuscript was well-written. In addition, this study has substantial clinical implications.

Major comments:

1-1. The authors need to provide information on the nature of the Shank3 mutation used and on how the mutation leads to impairments in the protein level/structure/function and synaptic structure/transmission/plasticity. It seems that the GGT to TTT mutation converts Gly to Phe in the PDZ domain of Shank3. Is this mutation from a patient? If not, what is the rationale for selecting this particular mutation over other mutations? How does it affect the structure and function/protein-interaction/stability of the PDZ domain; it would be easy to clarify this issue or perform simple modeling/experiments given the known crystal structure/PPIs of the Shank3 PDZ domain. Which splice variants of the Shank3 proteins are selectively affected?

The DNA sequence shown in Fig. 4c of the original manuscript (Supplementary Fig. 4a in the revised manuscript) is not a specific mutation sequence from ASD patients but a representative sequence. Owing to the nature of conventional Cas9-mediated non-homologous recombination, random mutations, either in-frame missense mutations or nonsense mutations, can be induced and vary between cells. We added Sanger sequencing data to Supplementary Figure 4b to provide more representative data exhibiting the frameshift mutation. The PDZ domain consists of the main Shank3 (Shank3a-Shank3d), which is expressed in the hippocampus (Wang et al. 2014). Therefore, by editing the PDZ domain of Shank3, the functions of the main Shank3 isoforms can be disrupted. We revised the manuscript to avoid further confusion.

Line 172-174:

We confirmed various mutations, including in-frame missense and nonsense mutations, induced in

the target sequence of *Shank3* by Sanger sequencing (Supplementary Fig. 4a, b) (...)

1-2. The study could also be strengthened by additional molecular/cell biological and electrophysiological data. Is the mutant Shank3 protein properly targeted to excitatory synapses, given that the Shank3 PDZ domain interacts with DLGAP/SAPAP proteins for synaptic localization? Does the decreased Shank3 puncta density in the vCA1 region accompany a decrease in the total or synaptic levels of Shank3, which can be tested by immunoblot analysis using Shank1/2/3 antibodies? What happens to the excitatory synaptic transmission (i.e., mEPSCs) in the Shank3-mutated CherryPicker-positive vCA1 neurons?

As mentioned in the following #1-1 question, we induced not a specific missense mutation but various missense and nonsense mutations in the PDZ domain, which led to a presumed reduction in the total levels of Shank3 protein, as depicted in Figure 4d. A validation study of the Shank3 antibody (Lutz et al. 2022) demonstrated that the anti-Shank3 antibody (Cell Signaling 64555S), which was used to stain Shank3 specifically in this study, detects the synaptic Shank3 protein and largely overlaps with other postsynaptic proteins. Therefore, we concluded that the reduction in immunostained puncta reflects the reduction of Shank3 at the synaptic level in Figure 4f due to the reduction in total Shank3.

Regarding excitatory synaptic transmission, although we appreciate your intriguing experimental proposal, it is difficult to perform *in vitro* patch clamp recordings specifically targeting Shank3-mutated CherryPicker-positive vCA1 neurons due to technical issues. In our experiment, we had to wait for five weeks for the degradation of endogenous Shank3 protein after EV injection. However, the CherryPicker protein, which was fused to Cas9 in the EV experiment to visualize the affected cells, was completely degraded within 5 weeks, as shown in Figure 4c. Therefore, targeting neurons for *in vitro* patch-clamp recording could not be achieved.

We have added a more detailed description of immunohistochemistry to the revised manuscript.

Line 175-177

We stained the Shank3 protein by immunohistochemistry using antibodies targeting synaptic Shank3 to further validate *Shank3* knockout efficiency at the protein level (Supplementary Fig. 4c, d). The knockout efficiency of *Shank3* could be quantitatively controlled by diluting the EV (Fig. 4e, f), which was consistent with the validation experiment with EGFP-targeting EV (Fig. 3d, e).

1-3. The authors carefully compare different dosages of Shank3-EVs and demonstrate that there

are threshold levels for the EVs. The amounts of total experiments that the authors performed are admirable, although not for Figure 4D. However, the vCA1 regions are rather large, and it is unclear whether the authors have targeted a large vCA1 region, or a small and particular vCA1 region, such as those shown in Figure 4, considering the unique projection patterns of the ipsilateral/contralateral CA2-CA1 circuits. These comments also apply to the results in Figure 1.

Thank you for your constructive comments. To further confirm this conclusion, we have added more samples to Figure 4g (Figure 4D in the original manuscript).

Our study targeted the pyramidal cell layer of the vCA1 and covered a relatively large vCA1 area. As mentioned in the following question (#1-2), the EV-injected position could not be detected 5 weeks after the EV injection in the mice where we performed the behavioral experiments. Therefore, we targeted the same region with the same injection protocol (same solution amount, speed, etc.) and confirmed the injection position 24 hours after surgery (Figure 4b, before the CherryPicker signal was degraded). We also targeted a relatively large area of vCA1 to target the pyramidal cell layer in the AAV experiments, as shown by the immunohistochemistry images of AAV-injected vCA1 (Fig. 1e). Thank you for your comments.

We have edited the manuscript to include the target of injection as follows:

Line 170-171

Shank3-cKO-EV was bilaterally injected into the hippocampal vCA1 region to target the pyramidal cell layer of vCA1 (Fig. 4a, b), (...)

Minor comments:

1-4. It is unclear what the authors mean by "actb normalized shank3 mRNA" in Figure 4D legend.

The quantified *Shank3* mRNA by RT-qPCR was divided by quantified beta-actin (*Actb*) mRNA from the same sample to normalize and demonstrate that the reduced *Shank3* mRNA was not due to the general reduction of transcribed mRNA.

We have changed the corresponding legend to improve clarity and added a more detailed protocol to the Materials and Methods section.

Original (Line 457)

(D) *actb* normalized *shank3* mRNA quantified by RT-qPCR.

Revised (Line 541-542)

d. *Shank3* mRNA normalized with beta-actin (*Actb*) mRNA quantified by qRT-PCR (see Materials and Methods).

Original (Line 373):

Light cycler 480 (Roche) is used following the standard protocol for absolute quantification (see Table S2 for primer sequences).

Revised (Line 455-458):

qRT-PCR was performed using the Thunderbird SYBR qPCR Mix (Toyobo) and Light Cycler 480 (Roche) following the standard protocol for absolute quantification (see Supplementary Table S2 for primer sequences). The quantified *Shank3* mRNA was divided by the quantified beta-actin (*Actb*) mRNA from the same sample for normalization.

Reviewer #2 (Remarks to the Author):

In this manuscript entitled “Conditional knockout of shank3 in the ventral CA1 by quantitative in vivo genome-editing impairs social memory,” the authors apply an in vitro developed vesicle gene editing system in a region specific manner to identify Shank3 expression in the vCA1 neurons as necessary for social memory. The results are novel and interesting in that they show that glutamatergic neurons in the vCA1 are a major contributor to social memory via synaptic mechanisms. The manuscript can be improved by addressing the following concerns:

Major:

2-1. Figure 1: it is important to show that Shank3 protein levels are reduced in this figure using immunostaining.

In a previous study (Guo et al. 2019), the DNA construct with the same sgRNA was validated. Specifically, the virus eliminated the major Shank3 isoforms in the region where the virus was injected. Furthermore, we performed puncta analysis in AAV-mediated knockout experiments (Supplementary Fig. 1a). We have included the data and description in the revised manuscript as follows:

Line 89-90:

We also confirmed the conditional knockout of Shank3 by immunohistochemistry for staining of the Shank3 protein (Supplementary Fig. 1a).

2-2. Figure 1E: can the authors provide some validation for the injection areas for vmPFC, NAcS, and LH as these are also included in the figure?

Thank you for the suggestion. We have added microscopic images of the injection areas for the vmPFC, NAcS, and LH in Supplementary Figure 1. We have also mentioned this in the revised manuscript as follows:

Line 96-99:

Next, to examine whether Shank3 functions in social memory in vCA1 downstream neurons, we injected the Shank3-cKO-AAV into major afferent projection regions of vCA1 neurons: the ventromedial prefrontal cortex (vmPFC), nucleus accumbens shell (NAcS), and lateral hypothalamus (LH) (Supplementary Fig. 1b-g).

2-3. Figure 1G: vCA1 interaction time is reduced for both novel and familiar mice, with the same amount of time as the familiar in the control group; meanwhile LH shows more interaction time similar to the amount of time spent with the novel animal in control. This suggests that LH region knock down more powerfully captures the social memory deficit and the mouse spends more time exploring both animals as novel. Can the authors clarify how to best interpret this discrepancy?

Thank you for pointing this out as an important point. We calculated the total social interaction time (Novel + Familiar) to examine whether there was a difference between groups. There was a slight trend, but no statistically significant difference, in the total interaction time between the groups (Supplementary Fig. 1h, One-way ANOVA, $F(4, 56) = 2.3$, $p = 0.07$). We have added the following descriptions to the manuscript and Supplementary Fig. 1h:

Line 101-102:

There was no difference in the total social interaction time during the social discrimination test between the groups (Supplementary Fig. 1h).

2-4. Figure 1: The authors claim that Shank3 expression in vCA1 “and its downstream regions” (line 100) are necessary for social memory behavior, but an effect is only shown with Shank3-cKO injection in the LH, and there are no studies showing the specific neurons that go from vCA1 to LH, as opposed to LH neurons receiving inputs from other brain regions, are responsible for the behavior. In order to make this claim, the authors would need to show a projection specific experiment.

Thank you for highlighting these important points. To answer your question, we additionally performed a projection-dependent conditional knockout of *Shank3* in the LH neurons that directly receive neural inputs from vCA1 (henceforth, $vCA1 \rightarrow LH$ neurons). AAV1-hSyn:Cre was injected into the vCA1 to achieve Cre-dependent anterograde transsynaptic labeling (Zingg et al. 2017). We injected a viral cocktail of AAV9-DIO-Cas9 and AAV9-U6:sgRNA/hSyn:DIO-EGFP into the LH to specifically label $vCA1 \rightarrow LH$ neurons with Cas9/sgRNA and EGFP. Interestingly, we did not observe a social memory deficit in the *Shank3* ^{$vCA1 \rightarrow LH$} -cKO mice, which indicates the LH neurons that receive direct inputs from vCA1 are not responsible for social memory. Based on these findings, we have added the following paragraphs:

Line 104-111

To investigate whether the Shank3 expression in the LH neurons which receive neural projection

from vCA1 (^{vCA1}→LH neurons) is essential for social memory, we performed projection pattern dependent *Shank3* conditional knockout (Supplementary Fig. 2a, b). Cre recombinase was anterogradely derived from vCA1 to the LH by transsynaptic delivery of AAV1-hSyn:Cre, and Cre-dependent Cas9 and sgRNA-expressing AAV were simultaneously injected into the LH (Supplementary Fig. 2a). Social memory dependent behavior was not disrupted by the *Shank3*-cKO in the ^{vCA1}→LH neurons, which suggests the social memory impairment observed in the *Shank3*-cKO in the LH neurons is not dependent on *Shank3*-cKO in the ^{vCA1}→LH neurons (Supplementary Fig. 2c, d).

Materials and Methods:

Line 323-331:

For AAV-mediated conditional knockout experiments, 300 nl of AAV9-CMV:Cas9-T2A-mCherry/U6:sgRNA (control or *Shank3*-targeting, 2×10^{12} genome copies [GC]/ml) was injected into each region. For the projection-dependent *Shank3* knockout experiment, 85 nl of AAV1-hSyn:Cre (Addgene, 2×10^{13}) was injected into vCA1 and 200 nl of a 1:1 cocktail of AAV9-pMeCP2:DIO-Cas9 (2×10^{12} GC/ml) and AAV9-U6:sgRNA(*Shank3*)/hSyn:DIO-EYFP (2×10^{12} GC/ml) (cKO) or AAV9-U6:sgRNA(*Shank3*)/hSyn:DIO-EYFP (2×10^{12} GC/ml) (Control) was injected into the LH. For *in vivo* EGFP-targeting EV-mediated genome-editing experiments, EGFP-targeting EV-#3 was injected into the NAc (+1.34 mm AP, ± 0.60 mm ML, and -4.60 mm DV). For the *in vivo* EV-mediated *Shank3*-cKO experiments, 500 nl of non-diluted and diluted *Shank3*-targeting EV was injected into bilateral vCA1.

Line 364-372:

The pAAV-pMeCP2:DIO-Cas9 plasmid was modified from pX551 (pAAV-pMeCP2:SpCas9-spA; Addgene plasmid # 60957) and was gifted by Dr. Dong Kong. The pAAV-U6:sgRNA/hSyn:mCherry and the pAAV-U6:sgRNA/hSyn:DIO-EYFP plasmids were modified from pX552 (pAAV-U6:sgRNA(SapI)/hSyn-GFP-KASH-bGH; Addgene plasmid # 60958). To make this modification, the original plasmid was digested using BamHI and EcoRI restriction enzymes. The PCR amplified mCherry sequence or the (BamHI)-DIO-EYFP-(EcoRI) fragment, which is a restriction enzyme digested product of pAAV-Ef1a:DIO-EYFP (Addgene plasmid # 27056), were then ligated using T4 DNA ligase (NEB) according to the standard protocol. *Shank3* targeting sgRNA (Supplementary Table 2, *Shank3*-sgRNA-2) was annealed and constructs were packaged in-house with AAV serotype 9.

2-5. Figure 2: The authors suggest that a certain number of neurons would be required to support a

social memory engram (line 102), however based on the above findings, one would expect that Shank3 expression in vCA1 > LH neurons are most important for this function. Can the authors clarify why they think the number of neurons, as opposed to type of neurons, or where the neurons project, is a relevant factor for vCA1 social memory function?

As previously mentioned, vCA1 contains various types of neurons with different neural projections and gene expression profiles (Gergues et al. 2020). We have also shown that social memory engram neurons in the vCA1 project to various brain regions and that among them, the neural circuit from the vCA1 to the NAcS has an essential function in social memory-based discriminatory behavior (Okuyama et al. 2016).

In contrast, in the current study, we did not specifically manipulate *Shank3* to target neurons from the perspective of cell type and neural projection. In AAV-mediated cKO, a nonspecific CMV promoter was used, whereas in EV-mediated cKO, nonspecific EVs with lipid bilayers were infected into the cells at the injection site. Taken together, it is expected that a certain number of NAcS-projecting vCA1 neurons are required to support a social memory engram. We described this in the Discussion section.

Line 250-251:

In this study, we exhibited a certain threshold of *Shank3*-unedited (i.e., *Shank3*-expressing) neurons, regardless of neural projection patterns, to show normal social memory function.

2-6. The authors claim that “AAV-mediated in vivo genome-editing may not be adequate to allow expression of the delivered gene in a particular cell proportion because of the difficulty in controlling infection and expression” (line 109), but do not show how dilution of EV compares to dilution of AAV. A comparison to AAV would justify the use/need of this technology better.

To directly compare the AAV- and EV-mediated cKO methods, we attempted to quantify the genome editing efficiency when diluting the AAV concentration. Specifically, we performed *in vitro* genome editing experiments with diluted AAV targeting the EGFP gene in the HEK293-EGFP cell line, similar to the experiments with diluted EV (Figure 3). We observed a decrease in the number of EGFP-negative cells after diluting the AAV solution, similar to that observed in the EV group. Importantly, we also observed a larger deviation (variable) in the proportion of EGFP-negative cells in the AAV group, presumably due to the infection conditions. We have added the results of the newly performed experiments to the manuscript as follows:

Line 143-151:

Next, to directly compare the efficiency of EV and AAV, we conducted a parallel experiment using AAV carrying Cas9 and EGFP targeting sgRNA (Fig. 2f, g, Supplementary Fig. 3d). Cas9 and EGFP targeting sgRNA encoding AAV vectors were packaged with AAV serotype 2, which has been previously reported to show high transduction efficiency in HEK-293 cells³². Upon diluting the EGFP-targeting AAV solution, we observed a reduction in the proportion of EGFP-negative cells three days after transduction, similar to the results obtained with EVs (Fig. 2g, Supplementary Fig. 3d). Notably, the AAV group exhibited a significantly higher variance than the EV group (Supplementary Fig. 3e, $p < 0.0001$, F-test). These findings suggest that EVs could serve as a viable alternative to AAV for quantitative *in vivo* genome editing, prompting our exploration of their application in investigating neural function.

2-7. Figure 3D,E: The authors specifying that 58% of cells in the NAcS express EGFP, but the data shown are expressed as Mean fluorescent intensity (Figure 3D) and relative ratio of mean intensity (Figure 3E). To make a conclusion about number of cells affected by knock down, a different measure would need to be used. Otherwise the conclusion statement should reflect this read out.

To quantitatively assess the number of cells affected by the conditional knockout, we counted and compared the number of EGFP-positive cells in the NAcS (Supplementary Fig. 3f-h). Although the proportion of EGFP-positive cells quantified in the NAcS (32.5%) was slightly lower than that reported in a previous study (Gong et al. 2003), presumably due to our setting of a strict threshold of fluorescence intensity to detect the most probable EGFP-positive cells, we found a decreased number of EGFP-positive cells in the EV-mediated cKO mice (Supplementary Fig. 3f-h). This corresponds with the mean fluorescence intensity analysis (Fig. 3d). We have included the relevant description in the Results section, and the counting methods have been added in the Materials and Methods section.

Line 159-164:

The EGFP fluorescent intensity and the number of EGFP-positive cells of the NAcS injected with EGFP-targeting EVs was significantly decreased, compared to those injected with the negative control EVs (Control, 16.5 ± 0.8 ; EGFP-targeting EV, 12.4 ± 1.2) (Fig. 3c, d, Supplementary Fig. 3f-h), indicating successful *in vivo* genome-editing via EV. In addition, similar to the *in vitro* experiment (Fig. 2e), the intensity of EGFP decreased in a concentration-dependent manner (Fig. 3e), suggesting the possible use of the system to control the proportion of the gene-edited cell population by diluting EV.

Line 437-443:

For the cell number analysis, microscopy images of EGFP-targeting EV-injected NAcS and control were cropped to 500 μm^2 using ImageJ. DAPI- and EGFP-positive cells were counted using the StarDist 2D plugin. The size and mean fluorescence intensity of the EGFP-positive region of interest (ROI) were measured on an 8-bit-converted EGFP fluorescence channel for the real dataset and on a different sample for the pseudo-dataset. Only cells within the size range of the mean \pm 1 SD and mean fluorescence above the threshold (67.614 out of a maximum of 255; the intersection point of histograms derived from real and pseudo datasets) were included for adjusted cell number.

2-8. Figure 4G: statistics performed are a series of paired t-test, but because the experiment involves two variables, 2 way ANOVA is required to determine statistical significance on dilution on behavior.

2-9. Figure 4H: Requires a one way ANOVA to say the dilution had an effect on behavior (similar to Figure 1H).

(2-8 & 2-9) In the revised manuscript, we performed a two-way repeated measures ANOVA on the data from Figure 4g. There were significant interaction effects between Familiarity and Dilution (Familiarity \times Dilution: $p = 0.0488$, $F(10, 99) = 1.937$). Additionally, we found a significant difference in the social discrimination score among the groups with different dilution levels, as determined by one-way ANOVA ($p = 0.0401$, $F(10, 99) = 2.009$, Fig. 4h). These findings support the notion that dilution affects social discriminatory behavior. We appreciate the suggestion to use appropriate statistical methods.

Line 546-548

g. Social discrimination test of Shank3-cKO-EV injected mice (1/1: $n = 9$; 1/2: $n = 10$; 1/4: $n = 12$; 1/8: $n = 11$; 1/16: $n = 5$; 1/32: $n = 6$; 1/64: $n = 5$; 1/128: $n = 15$; 1/256: $n = 11$; 1/512: $n = 12$; Control: $n = 14$, Two-way mixed model ANOVA).

2-10. Discussion: Line 187-188: authors state that a critical number of neurons are required to support social memory impairment, but Figure 4 does not provide any quantification of number of neurons affected.

As we mentioned in the Discussion, “...it was technically difficult to examine the exact percentage of Shank3-edited neurons in mice injected with diluted Shank3-cKO-EV... (Line XX)”, CRISPR-mediated gene editing irreversibly edits the target gene in the genome, allowing for a clear

classification of neurons as either edited or unedited. It exhibits distinctive features from siRNA- or antisense oligonucleotide-mediated knockdown methods, which only suppress the amount of expressed protein. Our *in vitro* (Figure 2) and *in vivo* (Figure 3) experiments with diluted EV targeting the GFP gene clearly showed a sequential decrease in the number of GFP-disrupted cells, which supports the idea that diluted EV targeting the Shank3 gene led to a sequential decrease in the number of *Shank3*-disrupted neurons. To quantify the number of edited neurons in the hippocampal vCA1 precisely, single-cell resolution analysis (e.g., single-cell RNA-seq) was essential, although it was not performed in our study. Based on available evidence, it can be considered that there is no change in the conclusion that a certain critical number of neurons is required for social memory function.

Minor:

2-11. Rather than going back and forth between “shank3” and “Shank3”, it may better to use “Shank3” referring to the murine gene throughout the manuscript.

Thank you for this suggestion. We changed all “*shank3*” that refers to the gene to “*Shank3*” (capitalized) throughout the manuscript.

2-12. Line 38: word missing after “The Shank3”

Thank you for the suggestion. We added "protein" after "The Shank3.”

Line 38:

"The Shank3 protein is expressed..."

2-13. Line 110: please clarify what is meant by “genomic scar”

We apologize for the confusion regarding terminology. In the original sentence, we intended to emphasize that the transient delivery of the Cas9 protein via EV could significantly lower the risk of off-target effects compared to the continuous expression of Cas9 by viral approaches (e.g., (Yin et al. 2016)). We modified the sentence as follows:

Original (Line 110):

In contrast, cell-derived EVs as natural carriers for DNA and other molecules have been proposed

to deliver molecules efficiently in a transient manner without leaving a genomic scar.

Revised (Line 120-122):

In contrast, cell-derived EVs have been proposed as natural carriers for DNA and other molecules to deliver molecules efficiently in a transient manner and significantly lower the risk of off-target effects.

2-14. Line 134: "the adult brains of EGFP-expressing transgenic mice", should be brains of adult EGFP-expressing transgenic mice"

Thank you for the correction. We changed the order of words as below.

Line 153-155:

Next, to examine whether EV-mediated *in vivo* genome-editing can be applied to brain tissue, we injected *EGFP*-targeting EVs into the brains of adult EGFP-expressing transgenic mice and quantified gene editing efficiency.

2-15. Line 213: "is complicated to" does not make sense in this sentence.

We appreciate the comment. We have modified the sentence as follows.

Original (Line 213):

Second, EV-mediated RNPs delivery is complicated to target a specific cell type.

Revised (Line 236-237):

Secondly, targeting specific cell types using EV-mediated RNPs delivery is challenging.

Reviewer #3 (Remarks to the Author):

This manuscript describes experiments in which conditional knockout of shank3 in the ventral CA1 region of the hippocampus was performed using quantitative *in vivo* gene editing. Effects on social memory were demonstrated. The manuscript is generally well written, however there are some major limitations. Comments below suggest ways in which the manuscript could be improved and extended.

General comments:

3-1. A direct comparison between transfection efficiencies achieved by AAV and EV methods would be desirable, since the authors state the superiority of the EV method.

In response to reviewer-2's question (#2-6), we conducted an *in vitro* direct comparison of the AAV and EV methods using the HEK293-EGFP cell line. We observed a similar trend of decreased numbers of EGFP-negative cells (i.e., edited cells) with higher variance when diluted AAV solution.

We have added the results of a newly performed experiment in the manuscript as below:

Line 143-151:

Next, to directly compare the efficiency of EV and AAV, we conducted a parallel experiment using AAV carrying Cas9 and *EGFP* targeting sgRNA (Fig. 2f, g, Supplementary Fig. 3d). Cas9 and *EGFP* targeting sgRNA encoding AAV vectors were packaged with AAV serotype 2, which has been previously reported to show high transduction efficiency in HEK-293 cells. Upon diluting the *EGFP*-targeting AAV solution, we observed a reduction in the proportion of *EGFP*-negative cells three days after transduction, similar to the results obtained with EVs (Fig. 2g, Supplementary Fig. 3d). Notably, the AAV group exhibited a significantly higher variance than the EV group (Supplementary Fig. 3e, $p < 0.0001$, F-test). These findings suggest that EVs could serve as a viable alternative to AAV for quantitative *in vivo* genome editing, prompting our exploration of their application in investigating neural function.

3-2. Dosage-dependency of social function on Shank3-KO EV dosage seems overstated by the authors. The data seems to show instead that there is a threshold – effects appear quite similar across undiluted EV dose to 1/64, only then is there a recovery of social function. The authors do conclude this according to Figure 4I; but the emphasis on dosage dependency prior to this does not

seem to be supported entirely by the data.

Thank you for pointing out this very important point. As mentioned in your comment, our data show that there is a dose-dependent threshold between a dilution factor of 1/64 and higher rather than a gradual loss/recovery of social memory. Additionally, in the revised manuscript, we have added the 1/256 and 1/512 diluted groups that exhibited normal social memory function comparable to that of the control group. These results supported the argument that a certain threshold exists. We revised the manuscript as follows:

Line 190-193:

Specifically, the non-diluted (1/1) group demonstrated no preference for novel mice compared with the 1/64 group, whereas the 1/128, 1/256, and 1/512 groups showed normal social discrimination behavior. Together, these results suggested that the threshold of *Shank3*-unedited neurons for obtaining normal social memory exists between 1/64 and 1/128 (Fig. 4i).

3-3. Higher quality microscopy images (especially at higher magnifications) would be good in Figures 3 and 4.

We replaced all microscopy images with higher-quality images. Thank you for the suggestion.

3-4. N numbers need to be explicitly listed in the manuscript. This omission, together with the fact that n numbers within the same experiment appear to vary widely between groups, give the impression that experiments may have been conducted by expanding certain groups without prior planning. The authors need to indicate all the n values for all groups in each graph/figure. Some n values look underpowered for mouse behavioral tests. Were power calculations performed before the experiments were started?

The number of individuals used for the behavioral experiments and statistical details are described in Supplementary Table 1. To decrease the variability of the N number among groups, we have added more N numbers and described them in the figure legends in the revised manuscript. Although power calculations were not performed in advance to determine the number of individuals required for the experiments, we followed the numbers of mice used in previous studies (e.g., (Okuyama et al. 2016)). Thank you for your comments.

3-5. The choice of statistical tests appears sometimes to be suboptimal, or incorrect. Why perform multiple t tests when there is the possibility to perform ANOVAs? Particularly relevant in the social interaction/recognition test.

In addition to Reviewer#2's comments (#2-8&2-9), we performed 2-way ANOVAs and One-way ANOVAs to compare the interaction time and social discrimination score of the social discrimination test, respectively, throughout the revised manuscript (Figures 1g, 1h, 2e, 2g, 3e, 4f, 4g, 4h). By a 2-way ANOVA, we found significant interactions between familiarity (familiar vs. novel) and treatment (Fig. 1g: AAV-cKO in different regions, $p = 0.0003$, $F(4, 56) = 6.27$; Fig. 4g: EV-cKO with serial dilution); $p = 0.0488$, $F(10, 99) = 1.937$). One-way ANOVA revealed significant differences between the groups (Fig. 1h, $p = 0.0005$, $F(4, 56) = 5.962$; Fig. 4h, $p = 0.04$, $F(10, 99) = 2.009$). Thank you for the suggestion.

3-6. The inconsistency between quantification methods chosen at different points (fluorescence intensity vs. shank3 puncta vs shank3 qRT-PCR) raises questions. The manuscript would benefit from being more consistent across the different parts.

To be more consistent across the experiments, in the revised manuscript, we have added a quantitative analysis of the number of EGFP-positive cells in the *in vivo* EGFP-targeting EV-injected mice (Supplementary Fig. 3f-h, related to Figure 3). Analyses of the fluorescence intensity shown in Figure 3d and e and the number of positive cells showed consistent results.

Regarding *Shank3* editing (Figure 4), since it was technically difficult to count the number of Shank3-expressing neurons by immunohistochemistry staining with the Shank3 antibody used in this study, we quantified the number of Shank3 puncta. We can now directly compare the editing efficiencies *in vivo* and *in vitro* experiments in the revised manuscript. Thank you for your constructive suggestions.

in vitro EGFP editing	Cell number counting (Fig. 2e)	
in vivo EGFP editing	Cell number counting (SupFig. 3f)	Fluorescence intensity (Fig. 3d, e)
	↓	
in vivo Shank3 editing	Puncta counting (Fig. 4f, SupFig. 4c, d)	qRT-PCR (Fig. 4d)

* Newly performed experiments in the revised manuscript are highlighted in red.

We have revised the manuscript as follows:

Line 159-162:

The EGFP fluorescent intensity and the number of EGFP-positive cells of the NAcS injected with *EGFP*-targeting EVs was significantly decreased, compared to those injected with the negative control EVs (Control, 16.5 ± 0.8 ; *EGFP*-targeting EV, 12.4 ± 1.2) (Fig. 3c, d, Supplementary Fig. 3f-h), indicating successful *in vivo* genome-editing via EV.

Line 175-178:

We stained the Shank3 protein by immunohistochemistry using antibodies targeting synaptic Shank3 to further validate *Shank3* knockout efficiency at the protein level (Supplementary Fig. 4c, d). The knockout efficiency of *Shank3* could be quantitatively controlled by diluting the EV (Fig. 4e, f), which was consistent with the validation experiment with EGFP-targeting EV (Fig. 3d, e).

3-7. Some important data in Figure 4 regarding the different dilutions is missing (see below for details).

We answered to your comments in the following part:

3-8. In Figure 1, this result is interesting (downstream targets), but seems like it is not explored further by the authors. An analogous experiment using EVs would be interesting, as it points towards wider circuitry involved beyond the ventral hippocampus. In Figure 1G, the authors use a paired t-test, which is inappropriate. Why not RM ANOVA? This would also give information on whether there is a statistical difference between the different injection conditions, not just whether their interaction levels with novel vs. familiar mouse are similar or different. E.g. NAcS injection seems to lead to lower differences between novel/familiar interactions.

Thank you for the suggestion. As we answered in Reviewer#2's comment (#2-4), we conducted neural projection-specific conditional knockout of *Shank3* in $vCA1 \rightarrow LH$ neurons to examine the relationship between vCA1 and its downstream regions in social memory function (Supplementary Figure 2). Our results showed that the conditional knockout of *Shank3* in LH neurons that received direct inputs from vCA1 did not affect social memory function.

In addition to comments #3-5 and Reviewer#2's comments (#2-8-9), we performed a 2-way ANOVA in Figure 1g (familiar vs. novel) and found that the vCA1-cKO and LH-cKO groups showed no difference in the interaction levels in the post-hoc analysis, whereas the control, vmPFC-, and NAcS-cKO groups showed a significant preference for novel mice (i.e., normal social memory). One-way ANOVA and post hoc analysis of the social discrimination score (Fig. 1h) further confirmed significant differences between the control, vCA-cKO, and LH-cKO groups.

3-9. In Figure 1 the authors should show the injection sites for every region.

Together with Reviewer 2's comment (#2-2), we have attached microscopy images of the injection sites in the new figure (Supplementary Fig. 1b-g). Thank you for the suggestion.

3-10. In Figure 1H, social discrimination score: As above, MWU test despite there being several groups? Why not ANOVA or non-parametric equivalent test (Friedman test)?

As we answered comments #3-5, we have performed a One-way ANOVA in the revised Fig. 1h instead of the MWU test. We found significant differences among the groups and have revised the manuscript accordingly (Fig. 1h, $p = 0.0005$, $F(4, 56) = 5.962$). Thank you for the suggestion.

3-11. In Figure 2E: Why are the individual datapoints not shown in this graph (despite being shown in 2F)? And "After 69 hours" – why was this timepoint chosen? Were other timepoints checked? Was a threshold reached at this point?

We added individual data points to Figure 2E. We apologize for this confusion. Regarding the phrase "After 69 hours," Considering the half-life of EGFP protein (15-26 hours) (Corish and Tyler-Smith 1999; Danhier et al. 2015), we have decided to check the genome editing efficiency at the timing of 72 hours (i.e., 3 days) of incubation after the EV transduction. We initially incubated the cells in the medium with the EV solution for 3 h. Subsequently, we replaced the medium with a normal cell culture medium and continued the incubation for an additional 69 h, resulting in 72 h of incubation after EV transduction. We have revised the Materials and Methods section as follows:

Line 402-405

The plates were centrifuged at 1,150 g for 30 min at RT and incubated at 37°C for at least 3 h initially. Subsequently, the medium was replaced to D-MEM without protamine sulfate, and continued incubation at 37°C for additional 69 h and prepared for flow cytometry analysis.

3-12. Figure 2E, n=1. The authors mentioned they repeated each experiment three times, but they need to be more clear whether it was n=1, repeated three times (technical replicates)?

As shown in Figure 2E (Supplementary Fig. 3b in the revised manuscript), we performed flow cytometry using two separate samples to confirm the presence of EGFP-targeting EV (#1–#4). There were no technical replicates of these samples. As you mentioned, we did not verify the reproducibility of the control group in the previous manuscript; therefore, we have included additional samples in the updated figure to address this concern. The updated figure also includes individual data points. Details have also been added to the Materials and Methods section.

Line 422-423:

Flow cytometry was performed on EV- or AAV-treated HEK293-EGFP cells for comparing the genome-editing efficiency of sgRNA (#1-#4) and the effects of serial dilution.

3-13. In Figure 2F: EV concentration: 100%, 50%, 25%, 12.5%, 1/28 and 6.25% - what concentration/amount was 100%? How was this achieved and quantified? If this 100% is the same as the one used for EV#3 in 1E; why is the proportion of EGFP negative cells much lower in this experiment (28 % vs. ~13%)? This is a stark difference and should be explained, otherwise it calls into question the consistency of the genome-editing efficiency between different experiments.

In this figure, the term "100%" refers to the undiluted EV solution after purification. To avoid confusion, we changed the notation for dilution from percentage to fraction. The differences in the proportion of EGFP-negative cells between EV#3 in Figure 2E (Supplementary Fig. 3c in the revised manuscript) and Figure 2F (Fig. 2e in the revised manuscript) panels were due to batch differences in the EV solution. Although we attempted to measure the concentration of EV in solution, we encountered difficulties using conventional methods such as flow cytometry, which cannot detect EVs owing to their small size. To maintain consistent and precise control of genome-editing efficiency, we used only a single batch of EV solution, as shown in Figures 3 and 4. We have added an explanation to the Materials and Methods section as follows:

Line 405-406:

In order to maintain consistent and precise control of genome-editing efficiency, we used a single batch of EV solution for each experiment.

3-14. In Figure 3: Negative control EVs - are these completely empty? Would have been good to also have a marker for these injections, as for the Shank-KO EVs. Was the location of control injections checked in a different way? If not, how can the authors be sure that they are looking/measuring in the exact area (and section) where the injection was at a comparable

concentration as the other side with Shank-KO EV?

The negative control EVs in our study (Figure 3 and Figure 4) are EVs that do not contain sgRNA but still contain Cas9 and CherryPicker proteins. The microscopic image in Figure 3b shows a representative unilateral injection used to demonstrate the injection site. In Figure 3c-e and Supplementary Figure 3f-h, the control EV were injected into the contralateral side of the EGFP-targeting EV-injected side (control side). As described in response to reviewer #1's comment (#1-3), the fluorescence 5 weeks after the injection site is not visible as the EV-conjugated CherryPicker protein degrades with the Cas9 protein (Fig. 4c). The same region was targeted using the same injection protocol (same solution amount, speed, etc.).

3-15. In Figure 3B: Image of Cherrypicker after 5 weeks is missing (text says that there was no Cherrypicker expression after 5 weeks, but there is no image). This should be included. Was DAPI also quantified? High magnification images of DAPI channel should also be shown and cell numbers quantified, otherwise there is the possibility that fluorescence may also be reduced due to dopaminergic cell death after EV injection rather than genome editing. Was only 1 section from 4 animals quantified? OR how was this done? Quantification procedure needs more detailed explanation in methods. Why is fluorescent intensity in %? Normally arbitrary units?

Thank you for your constructive suggestion. We included an image of an EV-injected mouse five weeks after injection (Fig. 4c). As stated in the manuscript, no Cherrypicker signals were detected at the injection sites. In response to Reviewer #2's comment (#2-7), we quantified the number of DAPI+ cells from the data used in Figure 3 and Supplementary Figure 3g but found no difference in the number of cells. To accomplish this, we used two slides from each individual and calculated the cropped image's intensity (0-255) by dividing it by the maximum value (255). We then convert this value into a percentage in the graph. Further details regarding this quantification method are provided in the Materials and Methods.

Original (Line 452):

For the EV-mediated genome-editing experiment, the EGFP (green) intensity of the nucleus accumbens shell of Drd1-EGFP was calculated by measuring the mean intensity of the selected area on RGB color-converted confocal microscopy images.

Edited (Line 430-443):

Microscopy analysis

For *in vivo* EV-mediated genome-editing experiments (related to Figure 3 and Supplementary

Figure 3), two slides of NAcS were collected from unilaterally EV-injected *Drd1*-EGFP mice. The fluorescence intensity values (0-255) were obtained using ImageJ (NIH) by measuring the mean intensity of the selected area on 8-bit-converted microscopy images. The values were then divided by the maximum intensity value (255) and expressed as a percentage. The relative ratio of the mean intensity was calculated by dividing the mean intensity (%) with that of the ipsilateral control side.

For the cell number analysis, microscopy images of EGFP-targeting EV-injected NAcS and control were cropped to 500 μm^2 using ImageJ. DAPI- and EGFP-positive cells were counted using the StarDist 2D plugin. The size and mean fluorescence intensity of the EGFP-positive region of interest (ROI) were measured on an 8-bit-converted EGFP fluorescence channel for the real dataset and on a different sample for the pseudo-dataset. Only cells within the size range of the mean \pm 1 SD and mean fluorescence above the threshold (67.614 out of a maximum of 255; the intersection point of histograms derived from real and pseudo datasets) were included for adjusted cell number.

3-16. In Figure 4D: qPCR *shank3*: how/to what is this normalised? The controls \sim 4.5; but then methods section mentions “absolute quantification”. This needs to be clarified either in Results or Methods section.

As we answered in response to Reviewer#1’s comment (#1-4), the quantified *Shank3* mRNA by the absolute quantification method was divided into the quantified *Actb* (beta-actin) mRNA from the same sample to normalize and demonstrate that the reduced *Shank3* mRNA was not due to the general reduction of transcribed mRNA. We have changed the corresponding legend to improve clarity and added a more detailed protocol to the Materials and Methods section.

Line 541-542:

d. *Shank3* mRNA normalized with beta-actin (*Actb*) mRNA quantified by qRT-PCR (see Materials and Methods).

Line 455-458:

qRT-PCR was performed using the Thunderbird SYBR qPCR Mix (Toyobo) and Light Cycler 480 (Roche) following the standard protocol for absolute quantification (see Supplementary Table S2 for primer sequences). The quantified *Shank3* mRNA was divided by the quantified beta-actin (*Actb*) mRNA from the same sample for normalization.

3-17. Furthermore, in Figure 4D the N numbers seem very low, especially control group. The

variability in controls is large, suggesting (for example via a power calculation) that larger n values are required.

Based on the comments from Reviewer#1 (#1-3), we have added more N to Figure 4d. Thank you for the suggestion.

3-18. In Figure 4D/4E, why not perform qPCR to quantify Shank3 at the different dilutions as well? In Figure 4E, these images give no information about the dilutions of Shank3-KO used, despite the text implying this. Which dilution are these images from? Where are these images taken in relation to the injection site? The Shank3-KO images seem to show that there is a gradient of expression - is this because the bottom right part is close to the injection site? If so, why was the injection site not focused on? The authors should show a lower magnification image (or drawing if unavailable) of the location.

We encountered difficulties in precisely quantifying the subtle differences in *Shank3* mRNA levels using qRT-PCR. Nevertheless, we found significant differences in the immunohistochemistry of Shank3 antibodies among control EV-injected WT, Shank3-cKO-EV injected WT, and conventional *Shank3*-KO mice. It has been reported that anti-Shank3 stained puncta reflects the actual Shank3 protein in the postsynaptic density (Lutz et al. 2022). Representative microscopic images of the injection sites were obtained from 1/64 and 1/128 EV-injected mice (Fig. 4e). We have also provided lower-magnification images of *Shank3*-KO mice in revised Figure 4e to demonstrate the exact region that we analyzed (the bottom-left part is the edge of the hippocampus). Thank you for your constructive comments. We revised the manuscript as follows:

Line 175-178:

We stained the Shank3 protein by immunohistochemistry using antibodies targeting synaptic Shank3 to further validate *Shank3* knockout efficiency at the protein level (Supplementary Fig. 4c, d). The knockout efficiency of *Shank3* could be quantitatively controlled by diluting the EV (Fig. 4e, f), which was consistent with the validation experiment with EGFP-targeting EV (Fig. 3d, e).

3-19. In Figure 4F: Why are the dilutions now referred to as 1/64 and 1/128 – how does this relate to the dilutions used previously and referred to as % values? The significance symbols do not show what is described in the figure legend. The comparison between 1/128 and controls should be denoted as significant, but instead the comparison between 1/128 and 1/64 is annotated with “*”. Puncta method of quantifying? Why switching between different methods of fluorescence

quantification? Also, what does one data point here represent? One individual? Or one section? In addition, it seems to me the most interesting comparison is missing – the diluted EVs vs. the undiluted EVs (or original dose). The authors make statements about the ability to quantitatively control the genome editing, but the data currently does not convincingly support that claim. In fact, it is surprising that a 1/128 dilution is still effective in reducing Shank3 expression vs. controls, and it begs the question as to how much the full dose reduces the expression in this type of quantification. Also, in the next paragraph, 5 further dilutions are used in injections before behavioral experiments. Were these injections also examined for their effect on Shank3 expression? Why are only the lowest ones shown?

We converted the percentages (50%, 25%) to fractions (1/2, 1/4) throughout the manuscript to improve readability. Regarding the significant symbols in Figure 4F, there is an error in the original figure. The corrected figure has been included in the revised manuscript (1/128 vs. control is significant; "*"). We appreciate your thorough review of our manuscript.

We quantified the number of puncta after immunohistochemistry of Shank3 in EV-injected brain sections to confirm the reduced Shank3 expression. As mentioned in the responses to #3-6 and #1-2, the Shank3 antibody we used accurately reflected the actual synaptic Shank3 expression (Lutz et al. 2022). Therefore, we could not quantify the number of Shank3 expressing cell, unlike in the case of EGFP-targeting EV experiments in *Drd1*-EGFP mice, where we measured both the fluorescence level and number of cells. Accordingly, we concluded that quantifying puncta is the optimal method for measuring reduced Shank3 expression. Additionally, we found a significant difference in the number of puncta in the vCA1 between Shank3 conventional knockout (*Shank3*-KO) and wild-type mice (Supplementary Fig. 4c, d). This supports the fact that the difference observed in the number of puncta (one section per point) at 1/64 and 1/128 accurately reflected the actual difference in expression levels. We selected these two dilutions to determine whether there were quantitative differences between the dilutions that showed behavioral differences. We have revised the manuscript as follows:

Line 137-140:

We performed the same assay using diluted EV solutions to examine the correlation between EV concentration and genome-editing efficiency. EVs with varying dilutions (dilution factors: 1/1, 1/2, 1/4, 1/8, and 1/16) and the negative control EVs containing only Cas9 protein were transduced into HEK293-EGFP cells, and the fluorescence was measured after three days.

3-20. In Figure 4G, behavior: N numbers seem on the low side for some groups, but then others have far larger samples sizes. Obviously, the statistical power achieved in the different conditions

is not comparable and the analyses with higher n numbers are able to detect smaller effect sizes. Additionally, the authors again only test for differences in interaction time between novel and familiar mice in each condition using a paired t test, whereas a repeated-measures ANOVA would be more appropriate, and differences between groups could be evaluated statistically. This result, arguably the main result of this study, is not very convincing in its current state. N numbers should be increased for most injection conditions to match 1/128 and control groups, and repeated-measures analysis should be performed.

In response to #3-4, we added more samples to the low-side group to diminish the variable N number in the same experiment. These new results were consistent and further confirmed the effect of serial dilutions of EV on social memory function. Together with the responses to #3-5, #2-8, and #2-9, we confirmed the effect of dilution through 2-way ANOVA (Fig. 4g) and found a significant interaction between dilution and familiarity ($p = 0.0488$, $F[10,99] = 1.937$) as well as a significant difference in social discrimination scores among the varied dilution groups by a One-way ANOVA ($p = 0.0401$, $F[10,99] = 2.009$, Figure 4h).

3-21. In the Methods: N numbers are missing, they should be explicitly stated somewhere, despite many graphs showing individual values, these are not clear enough.

The legend and Supplementary Table 1 indicate the number of samples for statistical details. Please refer to these sections for further statistical information.

3-22. Also in Methods: Social recognition test procedure needs clarifications: “demonstrators”/ “stimulators”? What is the role of “stimulators”? They are a different strain, and much younger? Should the mice not be strain and approximately age-matched?

We apologize for this confusion. To clarify, we used “demonstrators” throughout the revised manuscript for ‘familiarized’ and ‘novel’ mice in the social discrimination test. We chose juvenile male mice as demonstrators because age-matched mice often elicit aggressive behavior and can change the social context (Montagud-Romero et al. 2015). Additionally, following our previous report (Tao et al. 2022) we used two different strains of mice as demonstrators because mice discriminate better between different strains. We have added an explanation of the strain and age selection to the Materials and Methods section.

Line 300-306:

C3H/HeJ and BALB/c (5–8 weeks old; 20–25 g weighted) male mice were used for demonstrators. These mice were pre-handled for at least three days before the experiment. On the afternoon of the last day of habituation, a demonstrator mouse (either C3H/HeJ or BALB/c) was placed into the home cage of the subject mouse for familiarization (total duration 72 h). On the experimental day, a familiarized demonstrator mouse was separated for 30 min before behavioral recording. Familiar and novel demonstrator mice of a different strain (e.g., C3H/HeJ as a familiar demonstrator mouse, BALB/c as a novel demonstrator mouse and vice versa) were placed inside each mouse holder.

3-23. In the Methods, it is unclear: familiar mouse is C3H, but then on testing day the familiar mouse is BALB? When is BALB familiarised?

We familiarized the subject mice with a demonstrator mouse (C3H or BALBc, which were later used as familiar mice) for 72 h. On the test day, 30 min after separation, the familiarized demonstrator mouse (familiar mice) and novel demonstrator mouse of different strains (novel mice) were placed in the chamber simultaneously. The subject mouse had never seen a novel mouse previously. Along with comment#3-24, we revised the Materials and Methods section (please see the answer to #3-24).

3-24. Further Methods query: For 72 hours, C3H are placed in home cage – with the new mouse being so much younger, would this not lead to attacks by the resident mouse? How did the body weight compare between demonstrator mice and the experimental cohort? Were mice single housed or group housed throughout experiments?

Adult male mice exhibit less aggression towards juvenile male mice (Montagud-Romero et al. 2015). Based on these features, we used juvenile mice as demonstrators in our previous studies (Okuyama et al. 2016; Tao et al. 2022). Indeed, during our experiments, we observed no excessive aggression resulting in physical damage. C3H and BALBc body weights were controlled to be between 20-25 g, whereas subject mice (B6) were controlled to be between 25-30 g. The mice were group-housed throughout the experiments, except for a 30-minute separation period before the main test. We revised the Methods section as follows:

Line 296-306

The social discrimination test (SDT) protocol was modified from the original SDT. Briefly, 10–16 weeks old (25–30 g weighted) WT and *Shank3*-KO male mice were used for behavioral experiments. The subject mice were individually habituated to the experimenter over three days.

Habituation to the open field test chamber (380 mm × 380 mm × 300 mm) with two mouse holders (a quarter cylinder shape with a radius of 7 cm and height of 10 cm) on opposite corners was performed for 10 min each day. C3H/HeJ and BALB/c (5–8 weeks old; 20–25 g weighted) male mice were used for demonstrators. These mice were pre-handled for at least three days before the experiment. On the afternoon of the last day of habituation, a demonstrator mouse (either C3H/HeJ or BALB/c) was placed into the home cage of the subject mouse for familiarization (total duration 72 h). On the experimental day, a familiarized demonstrator mouse was separated for 30 min before behavioral recording. Familiar and novel demonstrator mice of a different strain (e.g., C3H/HeJ as a familiar demonstrator mouse, BALB/c as a novel demonstrator mouse and vice versa) were placed inside each mouse holder.

3-25. For immunohistochemistry and quantification procedures, the authors need to provide more detail. How many sections per mouse were quantified? Each mouse should still remain the experimental unit, i.e. if several sections are measured, statistics should be carried out on average values per mouse, not on values per section (shank3 puncta experiment).

Our study used 3-4 sections from each of the three individuals for immunohistochemistry. We have provided more detailed procedures in the Materials and Methods section, as follows:

Line 444-448

For Shank3 puncta analysis, three to four sections from each of three individuals were used for immunohistochemistry (Figure 4, Supplementary Figure 1, and Supplementary Figure 4). Areas including vCA1 pyramidal neurons were cropped and particles from the Shank3 fluorescence channel are analyzed and counted using ImageJ function “Analyze Particles” after setting a threshold. The number of particles was divided by the size of area.

Bibliography

- Corish, P., and C. Tyler-Smith. 1999. “Attenuation of Green Fluorescent Protein Half-Life in Mammalian Cells.” *Protein Engineering* 12 (12): 1035–40.
- Danhier, Pierre, Balaji Krishnamachary, Santosh Bharti, Samata Kakkad, Yelena Mironchik, and Zaver M. Bhujwala. 2015. “Combining Optical Reporter Proteins with Different Half-Lives to Detect Temporal Evolution of Hypoxia and Reoxygenation in Tumors.” *Neoplasia* 17 (12): 871–81.
- Gergues, Mark M., Kasey J. Han, Hye Sun Choi, Brandon Brown, Kelsey J. Clausing, Victoria S. Turner, Ilia D. Vainchtein, Anna V. Molofsky, and Mazen A. Kheirbek. 2020. “Circuit and Molecular Architecture of a Ventral Hippocampal Network.” *Nature Neuroscience* 23 (11): 1444–52.

- Gong, Shiaoqing, Chen Zheng, Martin L. Doughty, Kasia Losos, Nicholas Didkovsky, Uta B. Schambra, Norma J. Nowak, et al. 2003. "A Gene Expression Atlas of the Central Nervous System Based on Bacterial Artificial Chromosomes." *Nature* 425 (6961): 917–25.
- Guo, Baolin, Jing Chen, Qian Chen, Keke Ren, Dayun Feng, Honghui Mao, Han Yao, et al. 2019. "Anterior Cingulate Cortex Dysfunction Underlies Social Deficits in Shank3 Mutant Mice." *Nature Neuroscience* 22 (8): 1223–34.
- Lutz, Anne-Kathrin, Helen Friedericke Bauer, Valentin Ioannidis, Michael Schön, and Tobias M. Boeckers. 2022. "SHANK3 Antibody Validation: Differential Performance in Western Blotting, Immunocyto- and Immunohistochemistry." *Frontiers in Synaptic Neuroscience* 14 (June): 890231.
- Montagud-Romero, S., M. A. Aguilar, C. Maldonado, C. Manzanedo, J. Miñarro, and M. Rodríguez-Arias. 2015. "Acute Social Defeat Stress Increases the Conditioned Rewarding Effects of Cocaine in Adult but Not in Adolescent Mice." *Pharmacology, Biochemistry, and Behavior* 135 (August): 1–12.
- Okuyama, Teruhiro, Takashi Kitamura, Dheeraj S. Roy, Shigeyoshi Itoharu, and Susumu Tonegawa. 2016. "Ventral CA1 Neurons Store Social Memory." *Science* 353 (6307): 1536–41.
- Tao, Kentaro, Myung Chung, Akiyuki Watarai, Ziyang Huang, Mu Yun Wang, and Teruhiro Okuyama. 2022. "Disrupted Social Memory Ensembles in the Ventral Hippocampus Underlie Social Amnesia in Autism-Associated Shank3 Mutant Mice." *Molecular Psychiatry* 27 (4): 2095–2105.
- Wang, Xiaoming, Qiong Xu, Alexandra L. Bey, Yoonji Lee, and Yong-Hui Jiang. 2014. "Transcriptional and Functional Complexity of Shank3 Provides a Molecular Framework to Understand the Phenotypic Heterogeneity of SHANK3 Causing Autism and Shank3 Mutant Mice." *Molecular Autism* 5 (April): 30.
- Yin, Hao, Chun-Qing Song, Joseph R. Dorkin, Lihua J. Zhu, Yingxiang Li, Qiongqiong Wu, Angela Park, et al. 2016. "Therapeutic Genome Editing by Combined Viral and Non-Viral Delivery of CRISPR System Components in Vivo." *Nature Biotechnology* 34 (3): 328–33.
- Zingg, Brian, Xiao-Lin Chou, Zheng-Gang Zhang, Lukas Mesik, Feixue Liang, Huizhong Whit Tao, and Li I. Zhang. 2017. "AAV-Mediated Anterograde Transsynaptic Tagging: Mapping Corticocollicular Input-Defined Neural Pathways for Defense Behaviors." *Neuron* 93 (1): 33–47.

REVIEWER COMMENTS

Reviewer #1 (Remarks to the Author):

The authors fully addressed my review comments. I do not have additional comments.

Reviewer #2 (Remarks to the Author):

The revised manuscript by Chung and colleagues is significantly improved. The authors have responded to the specific critiques by adding new experiments and re-analyzing the data using appropriate statistical methods. In summary, the findings are more reliable and of interest.

Two minor comments:

1) Lines 102-104: Given the facts that LH-projecting neurons from vCA1 are not required for social memory and that vmPFC and NAcS specific KO of Shank3 does not result in social memory deficits, I suggest removing the following sentence: "Hence, these results suggest that Shank3 expression in vCA1 and its downstream regions is necessary for memory-dependent social discriminatory behavior."

2) There are still places where Shank3 gene is not italicized.

Reviewer #3 (Remarks to the Author):

The authors have extensively revised the manuscript and adequately addressed some, but not all, of my original comments. I detail below how specific comments still need to be fully addressed.

1. In their response to my original comment #4 (their 3-4 response) the authors state: 'To decrease the variability of the N number among groups, we have added more N numbers and described them in the figure legends in the revised manuscript.' The authors need to describe exactly how they 'added more N numbers'. They need to describe how the additional experiments were performed and what the n values were in the new experiments, how they ensured these were directly comparable with the original experiments, and the statistics behind pooling between different experiments.

2. In their response to my original comment #6 (their 3-6 response) the authors state 'Regarding Shank3 editing (Figure 4), since it was technically difficult to count the number of Shank3-expressing neurons by immunohistochemistry staining with the Shank3 antibody used in this study, we quantified the number of Shank3 puncta.' How can the authors be sure that these are Shank3-positive synapses? Many primary antibodies that appear specific on Westerns become less specific when used at a given concentration in an immunohistochemical protocol (where there is no way of determining whether the primary antibody is binding specifically to Shank3 and no other proteins).

3. In their response to my original comment #12 (their 3-12 response) the authors state: 'we have included additional samples in the updated figure to address this concern'. The authors need to describe exactly how they added more samples. They need to describe how the additional experiments were performed and what the n values were in the new experiments, how they ensured these were directly comparable with the original experiments, and the statistics behind pooling between different experiments.

4. In their response to my comment #13 in the original review (3-12 author response), the authors state 'To maintain consistent and precise control of genome editing efficiency, we used only a single batch of EV solution'. Does this impact negatively on the robustness and reproducibility of the

findings?

5. In their response (3-16) to my comment #16, the authors state that the control was beta-actin mRNA. Is this a sufficient control, or could beta-actin expression itself be altered in this experiment? Thus, is a further control required?

6. In response (3-17) to my comment #17, the authors state 'we have added more N to Figure 4d.' How were the additional experiments were performed, what were the n values were in the new experiments, how did they ensure these were directly comparable with the original experiments? They should also justify (statistically) the pooling between different experiments.

7. The response (3-18) to my comment #18 is unsatisfactory. The qPCR is needed. The 'Shank3 puncta' may not just be Shank3, but include puncta containing 'off-target' proteins that the antibody may also bind to in their particular immunohistochemical protocol.

8. The response (3-19) to my comment #19 is also insufficient, due to the doubts about the specificity, and accurate quantifiability of Shank3 immunoreactive puncta (in the absence of independent verification).

9. The response (3.20) to comment #20 states 'we added more samples to the low-side group to diminish the variable N number in the same experiment.' Once again, exactly how were these extra experiments carried out, were all groups included, and how was the pooling with the original data justified statistically?

9. The authors' response (3-25) to comment #25 includes 'Our study used 3-4 sections from each of the three individuals for immunohistochemistry'. This seems underpowered? Can the authors justify the statistical power in this and all related experiments?

Point-by-point response to the reviewer's comments

Reviewer #1 (Remarks to the Author):

The authors fully addressed my review comments. I do not have additional comments.

Thank you for carefully reading our manuscript and for your valuable comments.

Reviewer #2 (Remarks to the Author):

The revised manuscript by Chung and colleagues is significantly improved. The authors have responded to the specific critiques by adding new experiments and re-analyzing the data using appropriate statistical methods. In summary, the findings are more reliable and of interest.

Two minor comments:

1) Lines 102-104: Given the facts that LH-projecting neurons from vCA1 are not required for social memory and that vmPFC and NAcS specific KO of Shank3 does not result in social memory deficits, I suggest removing the following sentence: "Hence, these results suggest that Shank3 expression in vCA1 and its downstream regions is necessary for memory-dependent social discriminatory behavior."

We have removed the sentence from the manuscript as per your suggestion. Thank you for your feedback to improve the manuscript.

2) There are still places where Shank3 gene is not italicized.

Thank you for thoroughly reading our manuscript. We have italicized 'Shank3' to denote the gene throughout the manuscript and figures.

Reviewer #3 (Remarks to the Author):

The authors have extensively revised the manuscript and adequately addressed some, but not all, of my original comments. I detail below how specific comments still need to be fully addressed.

1. In their response to my original comment #4 (their 3-4 response) the authors state: 'To decrease the variability of the N number among groups, we have added more N numbers and described them in the figure legends in the revised manuscript.' The authors need to describe exactly how they 'added more N numbers'. They need to describe how the additional experiments were performed and what the n values were in the new experiments, how they ensured these were directly comparable with the original experiments, and the statistics behind pooling between different experiments.

Thank you for thoroughly reviewing our manuscript. Originally, we used a paired t-test to compare the differences in investigation time between familiar and unfamiliar mice. Accordingly, we determined the sample size necessary to achieve a power of 0.8 for the paired t-test (effect size = 1.754, $p = 0.05$). However, since we changed the analysis method to ANOVA according to reviewers' suggestion, we decided a current sample size to attain a power of 0.6 (effect size = 0.771, $p = 0.05$). The homogeneity of variance across all groups, between the original and new data, was confirmed using Levene's Test for Equality of Variances ($p > 0.05$, Table 1). The results of two-way mixed model ANOVAs show there is no significant difference between original and newly-added data (no significant interaction between the time of experiment performed and the investigation time toward novel and familiar mice, Table 2). All behavioral experiments in both the original and revised manuscript were performed in several batches, and we found the results to be consistent.

Group		Levene's statistic
AAV Control	Novel individuals: original vs. newly-added data	$F_{1,8} = 1.181, p = 0.309$
	Familiar individuals: original vs. newly-added data	$F_{1,8} = 0.448, p = 0.522$
vCA1-cKO	Novel individuals: original vs. newly-added data	$F_{1,8} = 0.806, p = 0.395$
	Familiar individuals: original vs. newly-added data	$F_{1,8} = 0.744, p = 0.414$
vmPFC-cKO	Novel individuals: original vs. newly-added data	$F_{1,8} = 0.056, p = 0.819$
	Familiar individuals: original vs. newly-added data	$F_{1,8} = 3.329, p = 0.105$
LH-cKO	Novel individuals: original vs. newly-added data	$F_{1,8} = 4.896, p = 0.094$
	Familiar individuals: original vs. newly-added data	$F_{1,8} = 1.795, p = 0.207$

Reviewer Table 1. Statistic details of homogeneity of variance assessed using Levene's Test for Equality of Variances for each group comparing the original and newly-added data.

Group	Interaction effect of the two-way mixed model ANOVA
AAV Control: original vs. newly-added data	$F_{1,8} = 2.046, p = 0.190$
vCA1-cKO: original vs. newly-added data	$F_{1,8} = 2.856, p = 0.130$
vmPFC-cKO: original vs. newly-added data	$F_{1,8} = 0.020, p = 0.892$
LH-cKO: original vs. newly-added data	$F_{1,11} = 0.478, p = 0.504$

Reviewer Table 2. Statistic details of the interaction effect using two-way mixed model ANOVA comparing original and newly-added data.

We have added this point in the manuscript as below:

Line 312-313

All the behavior experiments were performed in 3–5 batches and then combined.

2. In their response to my original comment #6 (their 3-6 response) the authors state 'Regarding Shank3 editing (Figure 4), since it was technically difficult to count the number of Shank3-expressing neurons by immunohistochemistry staining with the Shank3 antibody used in this study, we quantified the number of

Shank3 puncta.’ How can the authors be sure that these are Shank3-positive synapses? Many primary antibodies that appear specific on Westerns become less specific when used at a given concentration in an immunohistochemical protocol (where there is no way of determining whether the primary antibody is binding specifically to Shank3 and no other proteins).

Thank you for raising an important point. As shown in Supplementary Fig. 4c, d of the revised manuscript, the brain sections from homozygous whole-body knockout of *Shank3* (*Shank3*-KO) mice exhibit significantly fewer particles (puncta) using the current method outlined in the manuscript. Hence, we can infer that the number of immunohistochemically stained Shank3 puncta in our analysis corresponds to the amount of Shank3 protein.

Furthermore, as previously mentioned in our rebuttal letter, a previous study regarding the validation of Shank3 antibody clearly shows the Shank3 antibody used in our study successfully stains synaptic Shank3, as proven by co-immunohistochemistry staining with other synaptic molecules (Lutz et al., 2022).

Additionally, we conducted a new experiment (Supplementary Fig. 4e), examining the localization of Shank3 puncta in social memory engram neurons in the vCA1. The mixture of AAV-cfos:tTA and AAV-TRE-ChR2-EYFP was injected into the vCA1 and then a test-mouse socially interacted with a stimulator mouse after removing doxycycline (OFF-Dox period) to label social engram neurons with ChR2-EYFP. By staining for Shank3, we showed that the majority of Shank3 puncta overlapped with the spine-like structures of social memory engram neurons in the vCA1 (Supplementary Fig. 4e).

While we cannot completely eliminate the possibility of detecting few non-specific puncta, our results with *Shank3*-KO mice (Supplementary Fig. 4c, d) and new data with social memory engram neurons (Supplementary Fig. 4e), combined with the previous report on the antibody (Lutz et al., 2022), strongly support that puncta counting can be employed to detect reduced Shank3 expression in neurons. We have revised our manuscript accordingly:

Line 174-177:

We stained the Shank3 protein by immunohistochemistry using antibodies targeting synaptic Shank3 to further validate *Shank3* knockout efficiency at the protein level (Supplementary Fig. 4c-e). Shank3 puncta were detected in the dendritic spines of social memory engram neurons (Supplementary Fig. 4e).

Supplementary Fig. 4e

e. Shank3 puncta in the spine-like structure of the social memory neuron in vCA1. Left, Schematic illustration of the strategy of social memory neuron labeling; Right, Representative microscopic image of a social memory engram neuron stained with anti-GFP (green, EYFP) and anti-Shank3 (magenta, Shank3). White arrowhead, Shank3 puncta at the spine-like structure; Blue arrowhead, puncta in the dendrite. Bar = 2 μ m.

We have also added the related descriptions to the Materials and Methods in the revised manuscript.

Line 285-288:

For the social memory engram neuron labeling experiment, male C57BL/6J mice (7 weeks old) were fed with food containing 40 mg/kg doxycycline (Dox) for one week prior to surgery. The mice continued to be on Dox for the duration of the experiment, except on the neuron labeling day.

Line 335-336:

For the social memory engram neuron labeling experiment, 300 nl of 1:1 cocktail of AAV9-c-fos:tTA (2.0×10^{13} GC/ml) and AAV9-TRE:ChR2-EYFP (2.0×10^{13} GC/ml) was injected into bilateral vCA1.

Line 382-384:

The pAAV-c-fos:tTA and pAAV-TRE:ChR2-EYFP plasmids for the social memory engram labeling experiment were previously described (Roy et al., 2016). These plasmids were packaged in-house with AAV serotype 9.

Line 439-444:

Labeling of social memory engram neurons

72 hours before the social memory labeling experiment, the food containing Dox was replaced with normal food without Dox (OFF Dox). Age-matched, male WT (demonstrator) mice were then introduced into the home cage of the subject mice for a 2-hour social interaction. After the demonstrator mice were removed from the subject mice's home cage, the diet was switched back to food containing Dox (ON Dox). The subject mice were perfused 48 hours post-labeling.

Thank you for your comment.

3. In their response to my original comment #12 (their 3-12 response) the authors state: 'we have included additional samples in the updated figure to address this concern'. The authors need to describe exactly how they added more samples. They need to describe how the additional experiments were performed and what the n values were in the new experiments, how they ensured these were directly comparable with the original experiments, and the statistics behind pooling between different experiments.

There was no significant difference between the newly-added and the original data (independent sample t test, $t[1] = -8.561$, $p = 0.074$). Additionally, we found no difference between the combined data (Original + newly-added data) and new data ($p = 0.955$) as well as consistent statistics findings when compared with other groups (EV-#1-#4) using a One-way ANOVA followed by Tukey's multiple comparison test ($F[5,7] = 69.39$, $p < 0.0001$). Finally, we also obtained consistent results when comparing the newly-added data with EV-#1-#4, as shown in Supplementary Fig. 3c (One-way ANOVA followed by Dunnett's multiple comparison test, $F[4,5] = 90.1$, $p < 0.0001$). Therefore, we decided to pool the data for control group in the figure.

4. In their response to my comment #13 in the original review (3-12 author response), the authors state 'To maintain consistent and precise control of genome editing efficiency, we used only a single batch of EV solution'. Does this impact negatively on the robustness and reproducibility of the findings?

The results of behavioral experiment related to your question (Figure 4) were remarkably consistent across the experimental batches. We used EV solution from a single production in the present study due to the current methodological limitation of quantifying the number of Cas9-containing EVs. The advancement of measurement techniques for determining and controlling the concentration of EVs could address this issue in the future.

5. In their response (3-16) to my comment #16, the authors state that the control was beta-actin mRNA. Is this a sufficient control, or could beta-actin expression itself be altered in this experiment? Thus, is a further control required?

Beta-actin is widely used as control either for quantification of the amount of Shank3, both in Western blot analysis (Guo et al., 2019; Wang et al., 2014; Zhang et al., 2024) and in qRT-PCR (Zhang et al., 2024). We therefore used beta-actin mRNA as the control in our experiment. Following references are added in the new manuscript as follows:

Line 472-473:

The quantified Shank3 mRNA was divided by the quantified beta-actin (*Actb*) mRNA from the same sample for normalization as previously described (Zhang et al., 2024).

6. In response (3-17) to my comment #17, the authors state ‘we have added more N to Figure 4d.’ How were the additional experiments were performed, what were the n values were in the new experiments, how did they ensure these were directly comparable with the original experiments? They should also justify (statistically) the pooling between different experiments.

We decided the current sample size to attain a power of 0.8 for an independent t test (actual power = 0.816, effect size Cohen's $d = 0.784$, $p = 0.05$). The homogeneity of variance was met for both the blank EV group ($F[1,11] = 0.755$, $p = 0.404$) and the Shank3 EV group ($F[1,11] = 1.104$, $p = 0.316$) between the original and the newly-added data, confirmed by Levene's Test for Equality of Variances. The results of two-way ANOVAs show there is no significant difference between original and newly-added data ($F[1,22] = 0.598$, $p = 0.448$).

7. The response (3-18) to my comment #18 is unsatisfactory. The qPCR is needed. The ‘Shank3 puncta’ may not just be Shank3, but include puncta containing ‘off-target’ proteins that the antibody may also bind to in their particular immunohistochemical protocol.

8. The response (3-19) to my comment #19 is also insufficient, due to the doubts about the specificity, and accurate quantifiability of Shank3 immunoreactive puncta (in the absence of independent verification).

As we answered to your second comment, our results and prior research demonstrate that the Shank3 puncta effectively detects Shank3, and that quantitative counting of the number of puncta from immunohistochemically stained brain sections is sufficient for examining reduced Shank3 expression. We

concur that a qPCR experiment could potentially strengthen our results, but performing qPCR requires precise dissection of the area where AAV was microinjected. There's also the issue that the method of dissection can introduce variability into the data. While we appreciate your suggestion, our analysis of Shank3 puncta, which has demonstrated specificity in our results, is deemed sufficient to quantitatively examine the expression level of Shank3.

9. The response (3.20) to comment #20 states 'we added more samples to the low-side group to diminish the variable N number in the same experiment.' Once again, exactly how were these extra experiments carried out, were all groups included, and how was the pooling with the original data justified statistically?

The homogeneity of variance is met for the original and the newly added data (Levene's Test, $F[10,99] = 0.826$, $p = 0.605$), and there is no significant difference between the original and the new data addressed with two-way mixed-model ANOVA ($F[1,13] = 0.066$, $p = 0.802$). In addition, in the same vein with the answer for your third question, we have already pooled multiple behavioral analysis data from multiple batches due to the limit number of experiments one person can perform. This also significantly reduce the batch-effect of behavioral experiment simultaneously. We have added this point in the manuscript as below:

Line 312-313

All the behavior experiments were performed in 3–5 batches and then combined.

10. The authors' response (3-25) to comment #25 includes 'Our study used 3-4 sections from each of the three individuals for immunohistochemistry'. This seems underpowered? Can the authors justify the statistical power in this and all related experiments?

In Supplementary Fig. 4c and d, considering the variability among sections and individuals, we counted the number of puncta in pooled sections from multiple individuals. Indeed, we demonstrated the number of puncta in *Shank3*-KO were significantly lower in both pooled sections (Supplementary Fig., 4c) and individuals (Supplementary Fig. 4d).

References

- Guo, B., Chen, J., Chen, Q., Ren, K., Feng, D., Mao, H., Yao, H., Yang, J., Liu, H., Liu, Y., Jia, F., Qi, C., Lynn-Jones, T., Hu, H., Fu, Z., Feng, G., Wang, W., & Wu, S. (2019). Anterior cingulate cortex dysfunction underlies social deficits in Shank3 mutant mice. *Nature Neuroscience*, 22(8), 1223–1234.
- Lutz, A.-K., Bauer, H. F., Ioannidis, V., Schön, M., & Boeckers, T. M. (2022). SHANK3 Antibody Validation: Differential Performance in Western Blotting, Immunocyto- and Immunohistochemistry. *Frontiers in Synaptic Neuroscience*, 14, 890231.

- Wang, X., Xu, Q., Bey, A. L., Lee, Y., & Jiang, Y.-H. (2014). Transcriptional and functional complexity of Shank3 provides a molecular framework to understand the phenotypic heterogeneity of SHANK3 causing autism and Shank3 mutant mice. *Molecular Autism*, 5, 30.
- Zhang, H., Feng, Y., Si, Y., Lu, C., Wang, J., Wang, S., Li, L., Xie, W., Yue, Z., Yong, J., Dai, S., Zhang, L., & Li, X. (2024). Shank3 ameliorates neuronal injury after cerebral ischemia/reperfusion via inhibiting oxidative stress and inflammation. *Redox Biology*, 69, 102983.

REVIEWERS' COMMENTS

Reviewer #3 (Remarks to the Author):

The authors have further revised the manuscript and adequately addressed my comments. Whilst I think the qPCR data I suggested would strengthen the manuscript (as the authors acknowledge), their new data in their revised manuscript provides some reassurance.

Point-by-point response to the reviewer's comments

Reviewer #3 (Remarks to the Author):

The authors have further revised the manuscript and adequately addressed my comments. Whilst I think the qPCR data I suggested would strengthen the manuscript (as the authors acknowledge), their new data in their revised manuscript provides some reassurance.

Thank you for reading our manuscript thoroughly and for your valuable comments. We appreciate your feedback to improve the manuscript.